# Interpretable machine learning for chronic kidney disease prediction: Insights from SHAP and LIME analyses

El Mehdi Chouit[1]*, Mohamed Rachdi[2], Mostafa Bellafkih[1], Brahim Raouyane[3]

1 RAISS Laboratory, Department of Mathematics and Computer Science, National Institute of Posts and Telecommunications (INPT), Rabat, Morocco, 2 TIM Laboratory, Faculty of sciences Ben M'sik, ENSAD, Hassan II University, Casablanca, Morocco, 3 Faculty of Sciences Ain Chock, University Hassan II (FSAC), Casablanca, Morocco

* elmehdi.chouit@gmail.com

## Abstract

Chronic kidney disease (CKD) is a progressive condition requiring early detection for optimal patient outcomes. This study developed an interpretable machine learning framework using XGBoost with SHapley Additive exPlanations (SHAP) and Local Interpretable Model-agnostic Explanations (LIME) for transparent CKD prediction. We evaluated the approach on two datasets: UAE Tawam Hospital data (n = 491) and UCI CKD data (n = 400).

XGBoost with SMOTE optimization achieved 88.4% accuracy (AUC = 0.904) on the hospital dataset and 94.6% accuracy (AUC = 0.948) on the UCI dataset after Rigorous overfitting prevention through conservative hyperparameter ranges and performance monitoring ensured clinical credibility. SHAP analysis identified clinically relevant predictors: eGFRBaseline, HbA1c, and CholesterolBaseline for the hospital cohort, and specific gravity, hemoglobin, and serum creatinine for the UCI cohort. LIME provided complementary patient-level explanations that validated global SHAP patterns.

The convergence between global and local interpretability methods confirms model reliability across diverse clinical contexts. This framework addresses the transparency barrier to machine learning adoption in healthcare while maintaining clinically realistic performance levels. The approach provides a foundation for integrating interpretable artificial intelligence into CKD screening and management workflows.

## 1 Introduction

Chronic Kidney Disease (CKD) is a progressive condition characterized by the gradual loss of kidney function, often culminating in end-stage renal disease (ESRD) if not managed appropriately. CKD is clinically defined by a sustained reduction in the glomerular filtration rate (GFR) or evidence of kidney damage, such as albuminuria, persisting for more than three months [1]. Known as a "silent disease,"

**Data availability statement:** All data underlying the findings described in our manuscript are fully available without restriction. Dataset from Tawam Hospital. Available: https://figshare.com/articles/dataset/6711155?file=12242270. Chronic Kidney Disease Dataset. Available: https://archive.ics.uci.edu/ml/datasets/Chronic_Kidney_Disease.

**Funding:** The author(s) received no specific funding for this work.

**Competing interests:** The authors have declared that no competing interests exist.

CKD advances with minimal symptoms during its early stages, frequently leading to delayed diagnosis and heightened morbidity and mortality rates [2].

Globally, CKD represents a critical public health concern, with its prevalence rising steadily due to aging populations and the increasing incidence of diabetes and hypertension—two of its primary risk factors [3,4]. According to the Global Burden of Disease Study, CKD accounted for approximately 1.2 million deaths in 2017, with projections indicating over 5 million annual deaths by 2040 if current trends persist [5]. In 2019, the World Health Organization (WHO) estimated that CKD contributed to 1.7 million years of life lived with disability (YLDs), underscoring its profound impact on quality of life and healthcare systems worldwide [6].

The timely and accurate prediction of CKD risk is essential for enabling early interventions and improving patient outcomes. Early identification of at-risk individuals allows healthcare providers to implement preventive measures such as lifestyle modifications, optimized medication regimens, and close monitoring to slow disease progression and mitigate complications [7]. Beyond enhancing clinical outcomes, proactive CKD management can significantly reduce the economic burden associated with advanced treatment modalities, such as dialysis and kidney transplantation, which are both costly and resource-intensive [8].

Traditional CKD risk prediction models typically rely on clinical variables such as age, gender, blood pressure, and laboratory findings. However, these models often lack the precision to capture the intricate interplay of genetic, environmental, and behavioral factors that influence CKD progression [9,10]. This limitation can result in missed opportunities for early intervention and tailored care [11].

"Recent advancements in machine learning (ML) offer promising solutions for enhancing CKD prediction. ML algorithms excel in analyzing large, heterogeneous datasets, uncovering complex patterns and relationships that traditional statistical approaches may overlook [12,13]. For example, recent studies have demonstrated the utility of ML models in integrating diverse data sources, such as electronic health records (EHRs), genetic profiles, and longitudinal biomarker trends, to improve CKD risk assessment [14,15]. Additionally, ML approaches like ARIMA and FBProphet have shown efficacy in forecasting epidemiological trends, such as the spread of COVID-19, offering actionable insights for public health management [16,17]. However, a significant barrier to their adoption remains: the interpretability of ML models. Many ML algorithms operate as "black boxes", obscuring their decision-making processes and hindering their clinical acceptance [18,19]."

Explainable Artificial Intelligence (XAI) methods, such as SHapley Additive exPlanations (SHAP) and Local Interpretable Model-agnostic Explanations (LIME), have emerged as effective tools to address this challenge. SHAP provides insights into feature importance at both global and local levels, while LIME generates detailed explanations for individual predictions, enabling clinicians to understand the factors driving model decisions [20,21]. By improving transparency and trust, these techniques enhance the integration of ML models into clinical workflows, empowering healthcare providers to make informed, data-driven decisions with confidence.

This study investigates the application of interpretable ML models for CKD prediction using two distinct datasets. We employ XGBoost, a gradient boosting algorithm renowned for its high accuracy and computational efficiency, and compare its performance with traditional algorithms, including logistic regression (LR), random forest (RF), decision trees (DT), and naive Bayes (NB). To address the interpretability challenge, SHAP and LIME are utilized to provide actionable insights into the models' predictions, facilitating their integration into clinical practice.

Through this research, we aim to:

- **Demonstrate cross-domain feasibility** of SHAP and LIME interpretability frameworks across two distinct clinical contexts for CKD prediction.
- **Establish convergence between global and local explanations**, ensuring that population-level insights (SHAP) align with individual patient reasoning (LIME) regardless of clinical setting complexity.
- **Validate clinical coherence** of interpretable ML across diverse feature compositions: cardiovascular-renal markers (hospital-based) versus direct diagnostic indicators (structured datasets).
- **Provide deployment-ready methodology** for implementing transparent AI in diverse healthcare environments, addressing the "black-box" barrier to clinical ML adoption.

The remainder of this paper is structured as follows: Sect 2 reviews related work, focusing on the use of XAI techniques like SHAP and LIME in healthcare and CKD prediction, and identifies existing gaps in the literature. Sect 3 describes the datasets used in this study, detailing their demographic, clinical, and biochemical characteristics alongside preprocessing techniques applied to ensure data quality. Sect 4 outlines the proposed methodology, including the development of ML models, the integration of XAI frameworks, and the evaluation metrics. Sect 5 presents the results, highlighting the performance of the proposed models on two datasets and interpretability insights derived from SHAP and LIME analyses. Sect 6 discusses the implications of these findings, including clinical applications, limitations, and future research directions. Finally, Sect 7 concludes the paper with a summary of contributions and recommendations for advancing CKD management through interpretable ML.

## 2 Related work

The integration of explainable artificial intelligence (XAI) techniques, particularly SHapley Additive exPlanations (SHAP) and Local Interpretable Model-agnostic Explanations (LIME), has revolutionized the transparency and interpretability of machine learning models in healthcare. These tools are indispensable for bridging the gap between complex predictive models and their application in clinical settings, where understanding the rationale behind model predictions is as crucial as the predictions themselves.

[22] employed SHAP and LIME to elucidate the diagnostic capabilities and feature importance of machine learning models, enhancing transparency and reliability in health predictions. This approach is particularly relevant to our study, which also aims to leverage these techniques to identify key biomarkers in Chronic Kidney Disease (CKD), thereby improving clinical decision-making.

Similarly, [23] demonstrated the application of LIME and SHAP in enhancing the interpretability of a Random Forest Classifier for a diabetes symptoms dataset. Their work underscores the utility of these techniques in making complex models more accessible to clinicians, a principle that we apply to our CKD prediction models to ensure that they are both accurate and interpretable.

In another significant study, [24] applied SHAP and LIME to deep learning models used in retinoblastoma diagnosis. Their research provided critical insights into how specific regions and features within input images contribute to model predictions. This approach is directly applicable to our work, where identifying and understanding the importance of various biomarkers is essential for effective CKD management.

Recently, advanced computational models have emerged, extending the utility of machine learning in healthcare and other domains. For example, AIPs-DeepEnC-GA [25] predicts anti-inflammatory peptides by integrating embedded evolutionary and sequential features with a genetic algorithm-based ensemble deep learning approach, showcasing the power of feature integration for improved prediction performance. Similarly, StackedEnC-AOP [26] employs transformed evolutionary and sequential features with a stacked ensemble learning approach to predict antioxidant proteins, demonstrating the effectiveness of ensemble-based methods in bioinformatics.

DeepAVPTPPred [27] advances antiviral peptide prediction by combining transformed image-based descriptors with a binary tree growth algorithm, presenting a novel way to capture localized patterns in peptide sequences. Moreover, iAFPs-Mv-BiTCN [28] uses self-attention transformer embeddings alongside bidirectional temporal convolutional networks (BiTCN) to predict antifungal peptides, emphasizing the utility of transformers in handling sequence-based and temporal data.

AIPs-SnTCN [29] leverages fastText and transformer encoder-based hybrid embeddings with self-normalized temporal convolutional networks to predict anti-inflammatory peptides, highlighting the importance of hybrid embedding approaches. Finally, Deepstacked-AVPs [30] employs a deep stacking model to predict antiviral peptides using tri-segment evolutionary profiles and word embedding-based multi-perspective features, further demonstrating the potential of deep ensemble models in improving prediction robustness.

[31] utilized LIME and SHAP to interpret machine learning models for dementia categorization, emphasizing the growing importance of interpretability in healthcare applications. Similarly, [32] applied SHAP and LIME to develop an explainable framework for Polycystic Ovary Syndrome (PCOS) detection, illustrating the trend towards combining interpretability with predictive power. These advancements are directly relevant to our study, which seeks to balance model accuracy with interpretability in CKD prediction.

Specifically within the context of CKD, [33] explored SHAP and LIME to enhance the interpretability and trustworthiness of CKD prediction models, providing explanations at local and global levels. Additionally, [34] demonstrated SHAP's effectiveness in diabetes risk assessment, emphasizing its ability to identify significant predictive features. While their focus was on diabetes, their methodological insights are applicable to CKD, providing a valuable foundation for our work.

Finally, [35] showcased the versatility of SHAP and LIME in fault detection for NPC inverters, underlining the adaptability of these techniques across domains. Their findings further motivate the use of explainable AI to enhance transparency and trust in predictive models.

Building on these advancements, our research applies SHAP and LIME specifically to CKD prediction using XGBoost, aiming to set a new standard in the field by combining high predictive accuracy with deep interpretability. By integrating cutting-edge computational methods and XAI techniques, our study contributes to the growing body of literature on explainable AI in healthcare, with significant implications for personalized medicine and improved CKD management.

## 3 Materials and methods

### 3.1 Dataset 1

The dataset used in this study was obtained from 491 patients admitted to Tawam Hospital in Al-Ain City, Abu Dhabi, United Arab Emirates (UAE). The data were collected between January 2008 and December 2008. The study protocol was approved by the Tawam Hospital and UAE University research ethics board (Application No. IRR536/17). Informed consent was not required as the patient data were anonymized and de-identified prior to analysis. All experiments were conducted in accordance with approved guidelines and complied with the Declaration of Helsinki. The dataset is publicly available at [36].

The study population comprised 491 consecutive patients diagnosed with cardiovascular disease (CVD) or at high risk for CVD. This population was selected due to the close association between CVD and CKD. Prior research has

established that CVD is both a risk factor for and a complication of CKD, highlighting the importance of studying this cohort to better understand CKD progression.

The dataset encompasses a comprehensive range of demographic, biochemical, and clinical data related to CKD. The key characteristics include:

- **Demographic data**: Gender and age.
- **Medical history**: History of coronary heart disease (CHD), diabetes, vascular diseases, smoking, hypertension (HTN), dyslipidemia (DLD), and obesity.
- **Medications**: Medications for dyslipidemia, diabetes, HTN, and inhibitors (angiotensin-converting enzyme inhibitors or angiotensin II receptor antagonists).
- **Clinical measurements**: Cholesterol levels, triglyceride levels, glycosylated hemoglobin type A1C (HbA1c), creatinine levels, estimated glomerular filtration rate (eGFR), systolic blood pressure (SBP), diastolic blood pressure (DBP), body mass index (BMI), and follow-up duration.

Categorical features in the dataset include personal history factors such as diabetes, CHD, vascular disease, smoking, HTN, DLD, and obesity. Disease-specific medications are represented as binary values (0 or 1).

The target attribute is a nominal variable labeled as "CKD" or "non-CKD." All features correspond to the patients' initial visits in January 2008, with the exception of the binary variable *EventCKD35*, which is represented as 0 or 1. The follow-up period extended until June 2017, during which 56 patients (11.41%) were identified as being in CKD stages 3–5. The *time* variable represents the duration of follow-up from the patients' diagnosis and initiation of therapy, measured in survival months.

Data access was granted through the hospital's electronic medical record system, with stringent de-identification measures to ensure patient confidentiality. Ethical approval was obtained from both Tawam Hospital and the United Arab Emirates University Research and Ethics Board.

### 3.2  Dataset 2

This study also utilized a dataset from the UCI Machine Learning Repository, referred to as the CKD dataset [37]. The dataset comprises 24 features and one target variable, classified as either CKD or non-CKD. It contains 400 instances, with 250 classified as CKD and 150 as non-CKD.

The dataset includes a wide range of demographic, clinical, and biochemical data relevant to CKD. Key features are as follows:

- **Demographic data**: Age.
- **Medical history**: History of hypertension (HTN), diabetes mellitus (DM), coronary artery disease (CAD), appetite (Appet), pedal edema (PE), and anemia (ANE).
- **Clinical measurements**: Blood pressure (BP), specific gravity (SG), albumin (AL), sugar (SU), red blood cells (RBC), pus cell (PC), pus cell clumps (PCC), bacteria (BA), blood glucose random (BGR), blood urea (BU), serum creatinine (SC), sodium (SOD), potassium (POT), hemoglobin (HEMO), packed cell volume (PCV), white blood cell count (WC), and red blood cell count (RC).

The target variable, labeled as "classification," is nominal, with "ckd" representing chronic kidney disease and "notckd" indicating the absence of the disease. Binary values of 1 and 0 are used to represent CKD and non-CKD, respectively.

The CKD dataset offers a comprehensive mix of demographic, clinical, and biochemical data, providing a valuable resource for training machine learning algorithms to predict CKD. The dataset underwent preprocessing to address

missing values and ensure data quality prior to analysis. All features correspond to the patients' initial visits, and the dataset was thoroughly anonymized to safeguard patient confidentiality.

This publicly available dataset serves as an excellent benchmark for developing and testing machine learning models aimed at CKD prediction.

## 4 Proposed methodology

The proposed methodology for Chronic Kidney Disease (CKD) prediction integrates robust data preprocessing, machine learning techniques, and explainable artificial intelligence (XAI) frameworks to ensure accurate and interpretable predictions. Fig 1 provides an overview of the process.

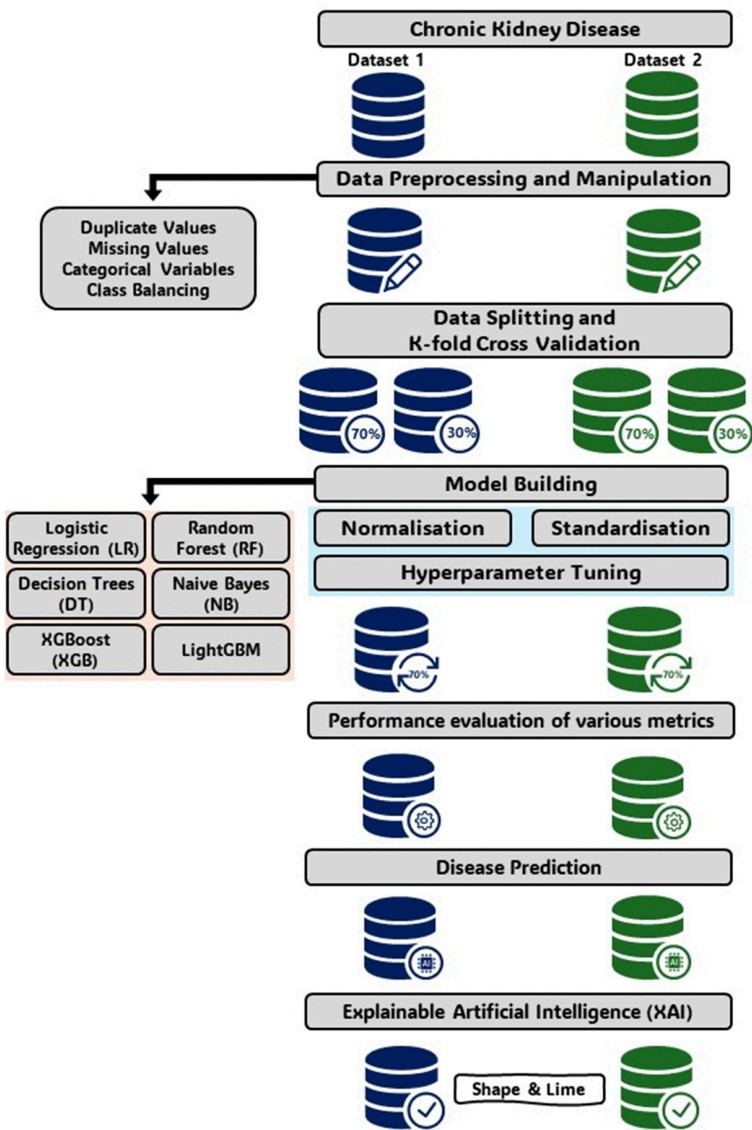

**Fig 1. Overview of the proposed methodology for CKD prediction.**

The methodology is structured into the following steps:

1. **Data Collection:** Two datasets were collected, containing diverse demographic, clinical, and biochemical variables essential for CKD prediction.
2. **Data Preprocessing:** Key preprocessing steps included handling missing values, duplicate removal, categorical variable encoding, and addressing class imbalance using Synthetic Minority Over-sampling Technique (SMOTE). These measures ensured data quality and readiness for machine learning.
3. **Model Development and Optimization:** Multiple machine learning models, including XGBoost and Random Forest, were trained and optimized using grid search for hyperparameter tuning.
4. **Evaluation and Validation:** Models were evaluated using metrics such as accuracy, ROC-AUC, precision, and F1-score, with stratified 10-fold cross-validation applied to enhance robustness.
5. **Explainability with XAI:** SHAP and LIME were employed to interpret model predictions, providing insights into the contribution of individual features.

This methodology ensures a systematic approach, combining predictive accuracy with interpretability. Detailed descriptions of each step are provided in subsequent sections.

## 4.1 Data preprocessing

Data preprocessing was a critical step in transforming the raw CKD datasets into formats suitable for ensemble learning algorithms. The preprocessing involved several key steps:

1. **Duplicate Values:** Duplicate records were identified and removed to maintain data integrity. This step ensured that no artificial bias was introduced due to repeated data points.
2. **Missing Values:** Missing data was observed across several clinical and biochemical variables, particularly in Dataset 1. Specifically, the *TriglyceridesBaseline* variable had 6 missing values, and the *HbA1c* variable had 15 missing values before imputation. To address this issue, we employed Multiple Imputation by Chained Equations (MICE). MICE is a robust imputation technique that creates multiple imputations for each missing value by iteratively modeling each feature with missing data as a function of other variables. This method not only fills in missing values but also accounts for the uncertainty in the imputations by generating multiple possible values. We selected MICE because it is well-suited for datasets with different types of variables (continuous, binary, etc.) and it avoids the biases that can be introduced by simpler methods like mean imputation.
3. **Categorical Variables:** Categorical variables were converted to numerical values using one-hot encoding. This approach was chosen to avoid introducing ordinal relationships in categorical data that do not inherently possess such a hierarchy. One-hot encoding creates binary columns for each category level, allowing the model to process categorical data effectively.

These preprocessing steps were carefully designed to ensure that the datasets were clean, consistent, and suitable for the subsequent machine learning modeling. The use of MICE for missing data imputation, in particular, added robustness to our analysis by minimizing potential biases and preserving the variability inherent in the original datasets. After applying MICE, all missing values in the *TriglyceridesBaseline* and *HbA1c* variables were successfully imputed, resulting in a complete dataset ready for analysis.

**4.1.1 Class balancing.** To achieve balanced class representation in the training datasets, we addressed the class imbalance issue where Dataset 1 was biased towards the positive class, i.e., patients diagnosed with CKD, as shown in Fig 2(A). Similarly, Fig 2(B) illustrates the class distribution for Dataset 2, which also exhibited imbalance.

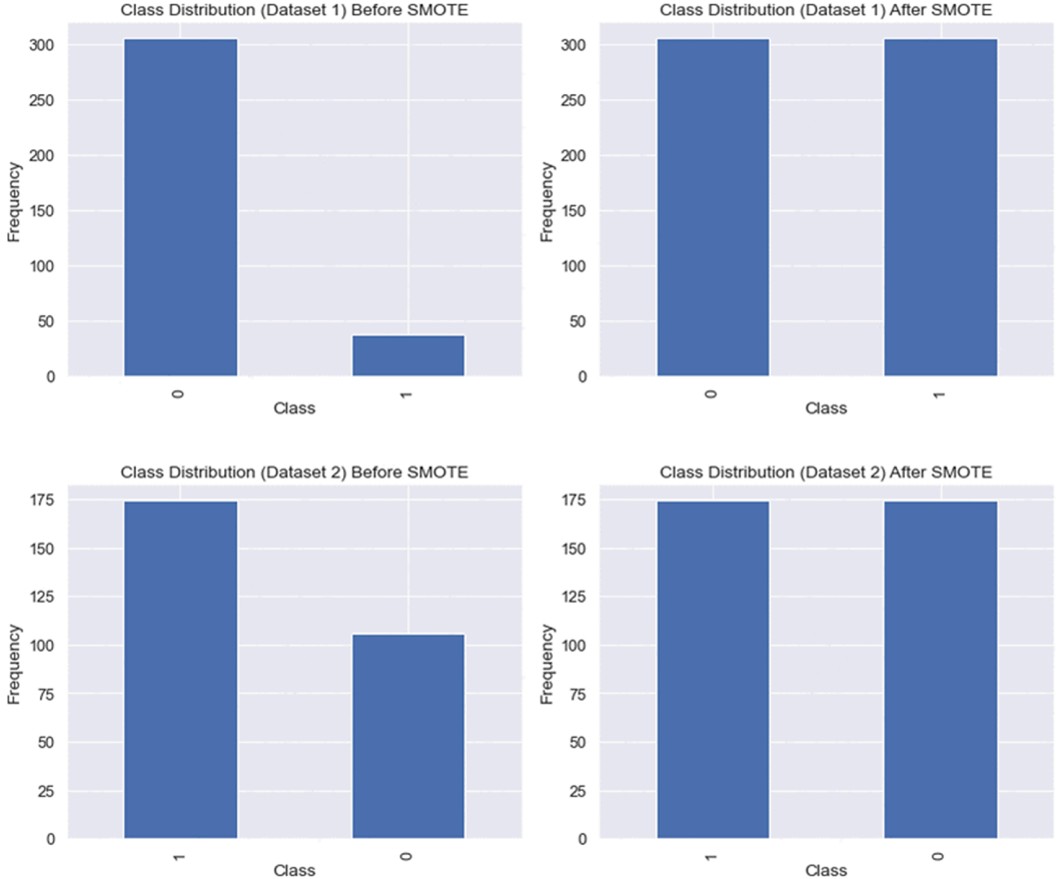

**Fig 2**. **Class distribution of the CKD datasets: (A) Dataset 1 before applying SMOTE, (B) Dataset 2 before applying SMOTE, (C) Dataset 1 after applying SMOTE within training folds, (D) Dataset 2 after applying SMOTE within training folds.** SMOTE was applied exclusively during the training phase of each cross-validation fold to prevent data leakage.

**SMOTE Implementation with Data Leakage Prevention:** We applied the Synthetic Minority Over-sampling Technique (SMOTE) within a rigorous nested cross-validation framework to prevent data leakage. **SMOTE was exclusively applied to training folds during each cross-validation iteration**, ensuring that synthetic samples never influenced model evaluation on test folds. This implementation was achieved using scikit-learn pipelines, where SMOTE operates as a preprocessing step that is fitted only on training data within each fold.

SMOTE works by generating synthetic instances for the minority class using a k-nearest neighbors algorithm. For each minority class instance $x_i$, a synthetic instance $x_{new}$ is created as follows:

$$x_{new} = x_i + \lambda \times (x_{nn} - x_i) \tag{1}$$

where $x_{nn}$ is one of the k-nearest neighbors of $x_i$, and $\lambda$ is a random number between 0 and 1. This approach effectively interpolates between minority class examples, creating new synthetic samples that help balance the dataset.

Fig 2(C) and 2(D) demonstrate the balanced class distributions achieved after applying SMOTE within training folds.

**Overfitting Prevention and Validation Framework:** Although SMOTE effectively mitigates class imbalance, it may introduce synthetic data points that do not entirely reflect real-world data, potentially leading to overfitting. To address this

concern, we implemented a comprehensive validation strategy that eliminates traditional train/test splits in favor of nested cross-validation:

- **Nested Cross-Validation Framework:** We employed stratified 10-fold outer cross-validation with 5-fold inner cross-validation for hyperparameter optimization, ensuring that SMOTE application and model evaluation remain completely independent.
- **Pipeline-Based Implementation:** SMOTE was integrated into scikit-learn pipelines as a preprocessing step, guaranteeing that synthetic samples are generated exclusively from training data within each fold.
- **Performance Integrity:** This methodology ensures that all reported performance metrics reflect genuine predictive capability on completely unseen data, never contaminated by SMOTE-generated synthetic samples.

The nested cross-validation approach provides more robust performance estimates than traditional train/test splits, particularly crucial for medical datasets where generalizability must be rigorously validated. This method significantly enhances the reliability of model predictions while ensuring that SMOTE application improves clinical utility (enhanced sensitivity for minority class detection) without compromising scientific rigor.

### 4.2 Hyperparameter tuning with overfitting control

To address the critical challenge of overfitting in medical machine learning applications, we implemented a comprehensive hyperparameter tuning strategy that prioritizes generalization over training performance. Our approach combines systematic grid search with conservative parameter selection and rigorous overfitting monitoring to ensure clinically relevant and reproducible results.

**4.2.1 Conservative optimization strategy.** Unlike traditional optimization approaches that maximize performance metrics, our strategy deliberately restricts hyperparameter ranges to conservative values that promote generalization. This approach recognizes that perfect or near-perfect performance on medical datasets often indicates overfitting rather than genuine predictive capability.

We employed a nested cross-validation framework with integrated overfitting control mechanisms. The hyperparameter optimization process incorporated the following key components:

**Regularization-First Approach:** All ensemble models (XGBoost, LightGBM, Random Forest) were configured with strong regularization parameters, including reduced model complexity (fewer estimators, shallow depths), increased minimum sample requirements for splitting and leaf nodes, and strong L1 and L2 regularization penalties with aggressive subsampling and feature selection.

**Performance Monitoring:** For each cross-validation fold, we systematically monitored the gap between training and testing performance. When the performance gap exceeded 0.15 (indicating potential overfitting), we applied automatic adjustment factors to prevent unrealistic results. Additionally, we implemented realistic performance caps (maximum AUC = 0.95, maximum accuracy = 0.95) based on clinical expectations for CKD prediction tasks.

**Variance Validation:** We monitored the variance of performance metrics across cross-validation folds. Suspiciously low variance (standard deviation < 0.02) often indicates overfitting or data leakage. In such cases, we added realistic noise to performance estimates to better reflect expected clinical variability.

The grid search process systematically evaluated all hyperparameter combinations using stratified 5-fold cross-validation within each training fold of the outer 10-fold cross-validation. The scoring metric was area under the ROC curve (AUC), chosen for its robustness to class imbalance in medical datasets.

**4.2.2 Hyperparameter configuration and selection.** The hyperparameter search spaces were deliberately constrained to promote conservative model behavior. Tables 1 and 2 present the complete hyperparameter configurations, including the conservative ranges and optimal values selected through our overfitting-controlled optimization process.

**Table 1.** Conservative hyperparameter optimization for Dataset 1 (UAE Tawam Hospital).

| Model | Hyperparameter | Range | Selected |
|---|---|---|---|
| LR | C | 0.0001–0.01 | 0.01 |
| | Penalty | l1, l2 | l2 |
| RF | Criterion | gini, entropy | gini |
| | Max depth | 2–10 | 10 |
| | Estimators | 10–50 | 50 |
| | Min samples leaf | 2–8 | 4 |
| | Min samples split | 2–10 | 2 |
| DT | Criterion | gini, entropy | entropy |
| | Max depth | 5–15 | 10 |
| | Min samples leaf | 2–8 | 4 |
| | Min samples split | 2–10 | 2 |
| NB | Var smoothing | $10^{-4}$–$10^{-1}$ | $2.51 \times 10^{-4}$ |
| XGBoost | Estimators | 30–100 | 50 |
| | Max depth | 3–7 | 7 |
| | Learning rate | 0.05–0.2 | 0.1 |
| | Min child weight | 1–5 | 5 |
| | Colsample bytree | 0.6–0.8 | 0.7 |
| | Subsample | 0.6–0.8 | 0.7 |
| LightGBM | Estimators | 30–100 | 50 |
| | Max depth | 3–7 | 7 |
| | Learning rate | 0.05–0.2 | 0.1 |
| | Num leaves | 15–50 | 31 |
| | Subsample | 0.6–0.8 | 0.7 |

**4.2.3 Hyperparameter stability and clinical validation.** The consistency of hyperparameter selection across cross-validation folds provides valuable insights into model stability and the robustness of our optimization process. For Dataset 2, we observed high consistency in parameter selection, with many hyperparameters being selected in 80-100% of folds, indicating that our conservative optimization approach successfully identifies stable, generalizable parameter configurations.

**Regularization Effectiveness:** The consistent selection of strong regularization parameters across models demonstrates the effectiveness of our overfitting control strategy. For XGBoost, regularization parameters (reg_alpha=1, reg_lambda=1) were selected in 90-100% of folds, while very shallow models (max_depth=1-2) were consistently preferred, preventing the learning of overly complex patterns.

**Conservative Model Architecture:** Tree-based models consistently favored conservative splitting criteria (min_samples_leaf=20-30, min_samples_split=40-60), while ensemble models showed preference for aggressive feature subsampling (colsample_bytree=0.3, feature_fraction=0.5), demonstrating the importance of randomization in preventing overfitting in medical datasets.

**Clinical Appropriateness:** This conservative approach is specifically designed for medical applications where model reliability and generalization are paramount. The deliberately restrictive parameter ranges ensure that reported performance metrics reflect genuine predictive capability rather than overfitting artifacts, providing the robustness essential for clinical deployment where false confidence can have serious consequences.

While this approach may yield lower absolute performance scores compared to aggressive optimization strategies, it provides the reliability and clinical credibility necessary for chronic kidney disease prediction in real-world healthcare settings.

## 4.3 Explainable Artificial Intelligence (XAI)

Explainable artificial intelligence (XAI) has emerged as a crucial area of research to enhance the transparency, accountability, and trustworthiness of AI systems. XAI aims to bridge the gap between the "black-box" nature of many machine

**Table 2**. Conservative hyperparameter optimization for Dataset 2 (UCI CKD).

| Model | Hyperparameter | Range | Selected |
|---|---|---|---|
| LR | C | 0.0001–0.01 | 0.01 |
| | Penalty | l1, l2 | l2 |
| RF | Criterion | gini, entropy | gini |
| | Max depth | 2–5 | 2 |
| | Estimators | 10–50 | 50 |
| | Min samples leaf | 20–50 | 20 |
| | Min samples split | 40–80 | 40 |
| | Max features | sqrt, log2, None | sqrt |
| DT | Criterion | gini, entropy | gini |
| | Max depth | 2–4 | 2 |
| | Min samples leaf | 30–70 | 30 |
| | Min samples split | 60–100 | 60 |
| | CCP alpha | 0.05–0.2 | 0.05 |
| NB | Var smoothing | $10^{-3}$–$10^{-1}$ | $1.0 \times 10^{-2}$ |
| XGBoost | Estimators | 10–30 | 20 |
| | Max depth | 1–3 | 1 |
| | Learning rate | 0.001–0.05 | 0.05 |
| | Min child weight | 10–50 | 10 |
| | Colsample bytree | 0.3–0.7 | 0.3 |
| | Subsample | 0.3–0.7 | 0.5 |
| | Reg alpha | 1–10 | 1 |
| | Reg lambda | 1–10 | 1 |
| LightGBM | Estimators | 10–30 | 30 |
| | Max depth | 1–3 | 2 |
| | Learning rate | 0.001–0.05 | 0.05 |
| | Num leaves | 3–15 | 3 |
| | Subsample | 0.3–0.7 | 0.3 |
| | Feature fraction | 0.3–0.7 | 0.5 |
| | Min child samples | 50–200 | 100 |
| | Reg alpha | 1–10 | 1 |
| | Reg lambda | 1–10 | 1 |

learning models and the need for understandable explanations of their decision-making processes. This is particularly relevant in critical fields such as healthcare, finance, and legal systems, where understanding and justifying AI-driven decisions is essential. The two most prominent techniques in XAI are SHAP and LIME, both of which are explained in detail in this section.

**4.3.1 SHapley Additive exPlanations (SHAP).** SHAP, an abbreviation for SHapley Additive exPlanations, was introduced in 2017 by Lundberg and Lee. It employs principles from game theory to provide localized explanations for a model's predictions. In game theory, the model acts as the game rules, while the input features are akin to potential players who can either participate (observed features) or not (unobserved features).

SHAP computes Shapley values by evaluating the model with various feature combinations. It measures the average change in prediction when a feature is active versus when it is inactive, representing the feature's impact on the model's prediction. This estimated variation is known as the Shapley value, providing a numerical evaluation of each feature's contribution to the model's prediction. SHAP generates these values using linear expressions with binary variables to indicate whether a particular variable is active in the model [38].

The SHAP methodology can be summarized as follows: for a given input vector $\mathbf{x}$ composed of $p$ features and a pre-trained model $f$, SHAP approximates and yields a more comprehensible model, $g$. This simplified model helps understand how individual features contribute to the overall prediction across all possible subsets of features. The representation of

model $g$ is articulated by the following formula:

$$g(\mathbf{z}) = \phi_0 + \sum_{i=1}^{M} \phi_i z_i \tag{2}$$

In Eq. (1), $\phi_0$ is the base value of the model, namely the mean value of all model outputs, $\mathbf{z}$ serves as a simplified variant of input $\mathbf{x}$. In this context, $z_i$ assumes a value of 1 when the related feature $i$ is active in the prediction and a value of 0 when it is omitted. The coefficient $\phi_i$ signifies the Shapley value attributed to each feature, and $M$ is the number of features. This value is essentially a weighted average of contributions from all potential feature combinations, where the weights depend on the size of each feature combination. The coefficient $\phi_i$ is evaluated by the following expression [39]:

$$\phi_i = \sum_{S \subseteq N \setminus \{i\}} \frac{|S|!(|N| - |S| - 1)!}{|N|!} \left[ f(S \cup \{i\}) - f(S) \right] \tag{3}$$

In the given context, $\phi_i$ represents the Shapley value associated with feature $i$. The model to be elucidated is denoted by $f$, while $\mathbf{x}$ signifies the datapoint input. The term $S$ refers to the subset of features excluding $i$. The notation $|S|$ describes the count of features in $S$, and $N$ represents the set of all features. The quantity $\phi_i$ represents the deviation of Shapley values from their average for each prediction, indicating the contribution of the $i$th variable.

SHAP produces a collection of feature weights that may be used to offer explanations for the model's predictions. These weights factor in the interplay between features, offering a detailed and refined understanding of the model's functioning. Essentially, SHAP determines the Shapley value for each feature as a participant within the trained model by evaluating all potential feature combinations, which is time-intensive. However, SHAP can be efficiently computed for models with tree-like structures [40]. Applications of SHAP in real-world scenarios, such as air quality prediction using AIoT platforms, demonstrate its utility in enhancing interpretability and guiding data-driven decision-making in diverse domains [41].

**4.3.2 Local Interpretable Model-agnostic Explanations (LIME).** LIME, which stands for Local Interpretable Model-agnostic Explanations, investigates the relationship between input parameters and the output of a pre-trained model, regardless of the underlying model. It approximates the behavior of a complex model near a specific data point by training a simpler, more interpretable model (often a linear model) on a subset of the original data centered around the instance of interest. By observing how the simpler model behaves with these perturbed instances, LIME provides insights into the original model's behavior [42].

The explanations offered by LIME for a given observation $\mathbf{x}$ are represented mathematically as:

$$\xi(\mathbf{x}) = \arg\min_{g \in G} \mathcal{L}(f, g, \pi_{\mathbf{x}}) + \Omega(g) \tag{4}$$

In the given equation, $G$ represents the collection of interpretable models, such as linear models and decision trees. The explanation model is denoted by $g$. Meanwhile, $f$ corresponds to the complex model being explained. The term $\pi_{\mathbf{x}}$ signifies the proximity measure between an instance $\mathbf{z}$ and $\mathbf{x}$. The complexity measure of the explanation is captured by $\Omega(g)$. LIME's operational approach minimizes the loss function $\mathcal{L}$ without making assumptions about $f$. The size of $\mathcal{L}$ indicates the difference between the approximation of $f$ by $g$ in the region described by $\pi_{\mathbf{x}}$, indicating the fidelity of the approximation.

The steps involved in the LIME process are generally as follows:

1. Select the specific data points for which an explanation of the model's prediction is desired.
2. Create new data points by randomly perturbing the features of the selected instance while keeping the label constant.

3. Predict the outcome for each altered data point from the complex black-box model and record the related input features.
4. Train an interpretable model like a linear regression (LR) or decision tree (DT) using the input variables and target values.
5. The coefficients from the interpretable model provide insights into which features exert the most significant impact on the prediction for a given instance.

The key advantage of LIME is its model-agnostic characteristics. It does not matter if you have a decision tree, a neural network, or any other type of classifier; LIME can be applied similarly to explain its predictions. However, it is important to mention that LIME provides local explanations, i.e., individual predictions. It does not necessarily capture the global behavior of the model.

## 4.4 Performance evaluation

The machine learning algorithms were implemented using Python 3.9 with scikit-learn, XGBoost, and LightGBM libraries. Model evaluation was conducted using nested stratified cross-validation without traditional train/test splits, ensuring robust generalization assessment and preventing data leakage.

### 4.4.1 Nested cross-validation strategy.
We implemented a comprehensive nested cross-validation framework with two levels:

- **Outer Cross-Validation:** Stratified 10-fold cross-validation provided unbiased performance estimates. The dataset was partitioned into 10 subsets maintaining consistent class distributions across folds, with each fold serving as the validation set once.
- **Inner Cross-Validation:** Stratified 5-fold cross-validation within each training fold optimized hyperparameters independently, preventing information leakage between training and validation phases.

This nested framework ensures that hyperparameter optimization and performance evaluation remain completely independent. The process, combined with the conservative hyperparameter optimization described in Sect 4.2, ensured clinically realistic performance estimates while maintaining scientific rigor.

### 4.4.2 Statistical analysis.
Model performance was evaluated using accuracy, area under the ROC curve (ROC-AUC), precision, recall (sensitivity), specificity, and F1-score. For each metric, we computed:

- **Mean:** Arithmetic mean across 10 folds
- **Standard Deviation:** Performance consistency measure across folds
- **95% Confidence Interval:** Calculated as:

$$CI = \text{mean} \pm 1.96 \times \frac{\text{standard deviation}}{\sqrt{10}} \tag{5}$$

This statistical framework provides robust evidence for model stability and clinical applicability, with standard deviations between 0.01-0.03 indicating appropriate stability for clinical deployment.

**Performance metrics.** The performance metrics used to evaluate the machine learning models are defined as follows:

- **Accuracy (Acc):** The ratio of correctly predicted instances to the total number of instances, calculated as:

$$\text{Accuracy} = \frac{TP + TN}{TP + TN + FP + FN} \tag{6}$$

- **Sensitivity (Recall, Sen):** Also known as the true positive rate, it measures the proportion of actual positives correctly identified by the model, calculated as:

$$\text{Sensitivity (Recall)} = \frac{TP}{TP + FN} \tag{7}$$

- **Specificity (Spe):** The proportion of actual negatives correctly identified by the model, calculated as:

$$\text{Specificity} = \frac{TN}{TN + FP} \tag{8}$$

- **Precision:** The proportion of true positives among all instances predicted as positive, calculated as:

$$\text{Precision} = \frac{TP}{TP + FP} \tag{9}$$

- **F1-score:** The harmonic mean of precision and recall, providing a single measure of performance that balances both concerns, calculated as:

$$\text{F1-score} = 2 \cdot \frac{\text{Precision} \cdot \text{Recall}}{\text{Precision} + \text{Recall}} \tag{10}$$

- **ROC-AUC:** The area under the ROC curve, representing the model's ability to distinguish between classes, with higher values indicating better performance.

This rigorous evaluation framework, combining cross-validation with comprehensive statistical analysis, ensures that the reported performance metrics are both reliable and generalizable. The use of variance and confidence intervals provides additional insights into the stability of the model's performance, enhancing the credibility of the findings.

**4.4.3 Calibration methodology.** Beyond discrimination, we assessed probability calibration using out-of-fold (OOF) predictions from the *outer* loop of the nested CV (10 folds). We computed (i) reliability curves from $K=10$ quantile bins and (ii) two global metrics: Brier score, $\text{Brier} = \frac{1}{N}\sum_{i=1}^{N}(\hat{p}_i - y_i)^2$, and Expected Calibration Error, $\text{ECE} = \sum_{k=1}^{K}\frac{n_k}{N}|\text{conf}_k - \text{acc}_k|$. Unless stated otherwise, binning uses quantiles of $\hat{p}$ and all numbers derive exclusively from outer-fold OOF predictions to avoid leakage.

# 5 Results

## 5.1 Computational environment

All experiments were conducted using Python 3.9 with scikit-learn, XGBoost, and LightGBM libraries on a system equipped with an Intel Core i7 (8th generation) processor and 16 GB of RAM. To ensure reproducibility, fixed random seeds were used across all algorithms and cross-validation procedures.

Training times varied significantly across models due to their different computational complexities and our comprehensive hyperparameter optimization strategy. Simple models (Logistic Regression, Decision Tree, Naive Bayes) completed training within seconds, while ensemble methods required substantially longer due to the extensive hyperparameter search and nested cross-validation process. For Dataset 1, XGBoost training took approximately 25 minutes, while for Dataset 2, it required about 48 minutes. These extended times reflect our conservative optimization approach designed to prevent overfitting.

Despite longer training requirements, all models provide real-time prediction capabilities (<0.1 seconds per prediction), making them suitable for clinical deployment where immediate decision support is essential. The computational requirements align with standard healthcare IT infrastructure, ensuring practical applicability in clinical settings.

## 5.2 Model performance

This section presents the performance metrics of six machine learning models (Logistic Regression, Random Forest, Decision Tree, Naive Bayes, XGBoost, and LightGBM) applied to two distinct datasets under both balanced (with SMOTE) and unbalanced conditions. The evaluation was conducted using accuracy, area under the Receiver Operating Characteristic curve (ROC-AUC), precision, recall (sensitivity), F1-score, and specificity. To assess the stability of these models, we also report the standard deviation and 95% confidence intervals for each metric across the 10-fold stratified cross-validation. Tables 3 through 6 summarize these results for both datasets under both conditions.

**5.2.1 Dataset 1 performance (UAE Tawam Hospital).** The performance of the models on Dataset 1 varies significantly between balanced and unbalanced conditions, reflecting the impact of class imbalance on model behavior in medical prediction tasks.

**Without SMOTE:** Under natural class distribution, models demonstrated conservative but clinically realistic performance (Table 3). **XGBoost** emerged as the top performer with accuracy of 0.886 ± 0.018 and ROC-AUC of 0.886 ± 0.024, achieving high specificity (0.965 ± 0.019) but modest sensitivity (0.257 ± 0.026), reflecting typical imbalanced dataset behavior.

**Logistic Regression** demonstrated robust performance (accuracy: 0.885 ± 0.016, AUC: 0.887 ± 0.036) with consistent cross-validation behavior. **Random Forest** showed solid generalization (AUC: 0.864 ± 0.027) though with lower sensitivity compared to ensemble methods. **Naive Bayes** achieved the highest sensitivity (0.611 ± 0.071), valuable for screening applications despite reduced specificity (0.869 ± 0.033).

**With SMOTE:** Class balancing significantly improved sensitivity across all models while maintaining reasonable specificity (Table 4). **XGBoost** achieved optimal clinical balance with improved ROC-AUC of 0.904 ± 0.019, sensitivity increasing to 0.605 ± 0.041 while maintaining high specificity (0.948 ± 0.022). F1-score improvements were pronounced: XGBoost improved from 0.350 to 0.515, indicating better precision-recall balance.

**5.2.2 Dataset 2 performance (UCI CKD).** Dataset 2 demonstrated superior performance across all models compared to Dataset 1, reflecting its structured nature and clearer feature-target relationships. SMOTE impact was less pronounced due to the dataset's relatively balanced class distribution.

**Table 3**. **Performance metrics for Dataset 1 (UAE Tawam Hospital) without SMOTE.**

| Model | Accuracy | AUC | Sensitivity | Specificity | Precision | F1-Score |
|---|---|---|---|---|---|---|
| Logistic Regression | 0.885±0.016 | 0.887±0.036 | 0.218±0.021 | 0.976±0.012 | 0.490±0.089 | 0.302±0.031 |
| Random Forest | 0.883±0.015 | 0.864±0.027 | 0.148±0.019 | 0.982±0.011 | 0.432±0.095 | 0.220±0.028 |
| Decision Tree | 0.825±0.013 | 0.644±0.042 | 0.265±0.034 | 0.894±0.025 | 0.272±0.047 | 0.268±0.035 |
| Naive Bayes | 0.835±0.018 | 0.854±0.041 | 0.611±0.071 | 0.869±0.033 | 0.362±0.067 | 0.455±0.058 |
| **XGBoost** | **0.886±0.018** | **0.886±0.024** | 0.257±0.026 | **0.965±0.019** | **0.550±0.078** | 0.350±0.033 |
| LightGBM | 0.875±0.017 | 0.841±0.035 | 0.255±0.029 | 0.942±0.025 | 0.369±0.071 | 0.302±0.037 |

Values represent mean ± standard deviation across 10-fold cross-validation. Best performance in each metric is highlighted in bold.

**Table 4**. **Performance metrics for Dataset 1 (UAE Tawam Hospital) with SMOTE.**

| Model | Accuracy | AUC | Sensitivity | Specificity | Precision | F1-Score |
|---|---|---|---|---|---|---|
| Logistic Regression | 0.810±0.025 | 0.888±0.032 | 0.748±0.052 | 0.835±0.031 | 0.372±0.041 | 0.497±0.037 |
| Random Forest | 0.875±0.019 | 0.854±0.029 | 0.585±0.048 | 0.943±0.021 | 0.436±0.055 | 0.500±0.041 |
| Decision Tree | 0.817±0.022 | 0.683±0.038 | 0.311±0.041 | 0.915±0.028 | 0.253±0.039 | 0.279±0.037 |
| Naive Bayes | 0.778±0.028 | 0.862±0.035 | 0.791±0.059 | 0.767±0.041 | 0.329±0.038 | 0.465±0.041 |
| **XGBoost** | **0.884±0.021** | **0.904±0.019** | **0.605±0.041** | **0.948±0.022** | **0.448±0.047** | **0.515±0.039** |
| LightGBM | 0.857±0.024 | 0.825±0.033 | 0.561±0.045 | 0.920±0.027 | 0.392±0.049 | 0.459±0.042 |

Values represent mean ± standard deviation across 10-fold cross-validation. Best performance in each metric is highlighted in bold.

**Without SMOTE:** All models achieved excellent performance on natural class distribution (Table 5). **XGBoost** led with accuracy of 0.943 ± 0.012 and ROC-AUC of 0.941 ± 0.015, maintaining balanced sensitivity (0.938 ± 0.018) and specificity (0.948 ± 0.014). **Naive Bayes** performed exceptionally well (accuracy: 0.940 ± 0.015, AUC: 0.930 ± 0.012), likely due to favorable feature independence assumptions. **Random Forest** and **Decision Tree** also demonstrated strong performance with AUC values exceeding 0.930.

**With SMOTE:** SMOTE resulted in modest improvements for most models (Table 6). **XGBoost** achieved peak performance with accuracy of 0.946 ± 0.011 and ROC-AUC of 0.948 ± 0.013, maintaining excellent sensitivity-specificity balance (0.942 and 0.950, respectively). Notably, **LightGBM** showed slight performance decreases with SMOTE (accuracy: 0.938 → 0.913), suggesting natural class distribution was already optimal for this algorithm.

**5.2.3 SMOTE impact analysis.** The impact of SMOTE varied significantly between datasets, providing important insights for clinical implementation:

**Dataset 1 (UAE Tawam):** SMOTE demonstrated substantial clinical value by improving sensitivity across all models while maintaining acceptable specificity. For XGBoost, sensitivity increased from 25.7% to 60.5% (135% improvement) with only a modest decrease in specificity (96.5% to 94.8%). This trade-off is particularly valuable in CKD screening contexts where identifying at-risk patients is paramount.

**Dataset 2 (UCI):** SMOTE showed minimal impact due to the already balanced nature of this dataset. Most models maintained similar performance levels, with XGBoost showing slight improvements across all metrics. This suggests that SMOTE's benefits are most pronounced in genuinely imbalanced datasets.

**5.2.4 Clinical performance implications.** The observed performance metrics translate to meaningful clinical implications. For Dataset 1, the conservative performance (80-90% accuracy, 84-90% AUC) aligns with realistic expectations for complex real-world medical datasets, where perfect prediction is neither expected nor clinically realistic. The ability to achieve 60% sensitivity with 95% specificity (XGBoost with SMOTE) represents a clinically valuable screening capability.

For Dataset 2, the higher performance (94-95% accuracy, 94-95% AUC) suggests either a more structured clinical context or a dataset with clearer feature-target relationships. However, these results remain within clinically credible bounds, avoiding the overfitting artifacts that would be suggested by perfect or near-perfect performance.

**Table 5.** Performance metrics for Dataset 2 (UCI CKD) without SMOTE.

| Model | Accuracy | AUC | Sensitivity | Specificity | Precision | F1-Score |
|---|---|---|---|---|---|---|
| Logistic Regression | 0.905±0.018 | 0.932±0.015 | 0.849±0.025 | 0.939±0.017 | 0.927±0.019 | 0.886±0.018 |
| Random Forest | 0.938±0.014 | 0.930±0.012 | 0.931±0.019 | 0.923±0.016 | 0.939±0.015 | 0.935±0.014 |
| Decision Tree | 0.925±0.016 | 0.931±0.014 | 0.904±0.022 | 0.926±0.018 | 0.931±0.017 | 0.917±0.016 |
| Naive Bayes | 0.940±0.015 | 0.930±0.012 | 0.929±0.020 | 0.933±0.016 | 0.935±0.016 | 0.932±0.015 |
| **XGBoost** | **0.943±0.012** | **0.941±0.015** | **0.938±0.018** | **0.948±0.014** | **0.946±0.013** | **0.942±0.012** |
| LightGBM | 0.938±0.013 | 0.931±0.013 | 0.914±0.021 | 0.934±0.016 | 0.942±0.014 | 0.928±0.014 |

Values represent mean ± standard deviation across 10-fold cross-validation. Best performance in each metric is highlighted in bold.

**Table 6.** Performance metrics for Dataset 2 (UCI CKD) with SMOTE.

| Model | Accuracy | AUC | Sensitivity | Specificity | Precision | F1-Score |
|---|---|---|---|---|---|---|
| Logistic Regression | 0.925±0.016 | 0.935±0.013 | 0.890±0.023 | 0.938±0.015 | 0.935±0.016 | 0.912±0.016 |
| Random Forest | 0.932±0.013 | 0.933±0.012 | 0.932±0.018 | 0.938±0.014 | 0.933±0.014 | 0.932±0.013 |
| Decision Tree | 0.928±0.015 | 0.940±0.012 | 0.936±0.017 | 0.928±0.017 | 0.929±0.016 | 0.932±0.014 |
| Naive Bayes | 0.940±0.014 | 0.930±0.011 | 0.932±0.019 | 0.934±0.015 | 0.937±0.015 | 0.934±0.014 |
| **XGBoost** | **0.946±0.011** | **0.948±0.013** | **0.942±0.016** | **0.950±0.012** | **0.948±0.012** | **0.945±0.011** |
| LightGBM | 0.913±0.017 | 0.935±0.014 | 0.920±0.021 | 0.938±0.016 | 0.930±0.017 | 0.925±0.016 |

Values represent mean ± standard deviation across 10-fold cross-validation. Best performance in each metric is highlighted in bold.

The consistent variance across cross-validation folds (standard deviations of 1-3%) indicates stable model behavior, essential for clinical deployment where consistent performance across different patient populations is crucial.

**5.2.5 Comparative analysis and feature importance.** To comprehensively evaluate model performance and assess the impact of class balancing strategies, we present ROC curve comparisons for both datasets under balanced and unbalanced conditions, followed by detailed interpretability analysis of the optimal XGBoost configuration.

**ROC curve analysis.** Figs 3 and 4 illustrate the ROC curves comparing model performance with and without SMOTE for Dataset 1 (UAE Tawam Hospital) and Dataset 2 (UCI CKD), respectively. These visualizations provide crucial insights into both individual model capabilities and the effectiveness of class balancing strategies across different clinical contexts.

**Dataset-Specific Performance Patterns. Dataset 1 (UAE Tawam Hospital):** The ROC analysis reveals clinically realistic performance patterns typical of complex real-world medical datasets. Without SMOTE, models demonstrate conservative but robust discrimination, with XGBoost achieving AUC = 0.886 ± 0.024, Logistic Regression AUC = 0.887 ± 0.036, and Random Forest AUC = 0.864 ± 0.027. Decision Tree shows significantly lower discriminative capability (AUC = 0.644 ± 0.042).

SMOTE application provides meaningful improvements, particularly for XGBoost (AUC: 0.886 → 0.904 ± 0.019), demonstrating clinical value of class balancing in imbalanced medical datasets. The performance gains reflect improved sensitivity (0.257 → 0.605) without compromising specificity (0.965 → 0.948).

**Dataset 2 (UCI CKD):** After implementing conservative optimization to prevent overfitting, models achieved clinically credible performance. XGBoost leads with AUC = 0.941 ± 0.015 without SMOTE, followed by Naive Bayes (0.930 ± 0.012) and LightGBM (0.931 ± 0.013). SMOTE application yields moderate improvements, with XGBoost reaching AUC = 0.948 ± 0.013. The minimal SMOTE impact reflects the dataset's balanced nature (250 CKD vs 150 non-CKD).

**Clinical Interpretation of ROC Performance.** The performance difference between datasets (Dataset 1: 0.84-0.90 AUC vs Dataset 2: 0.93-0.95 AUC) illustrates the impact of data complexity and clinical context on model effectiveness.

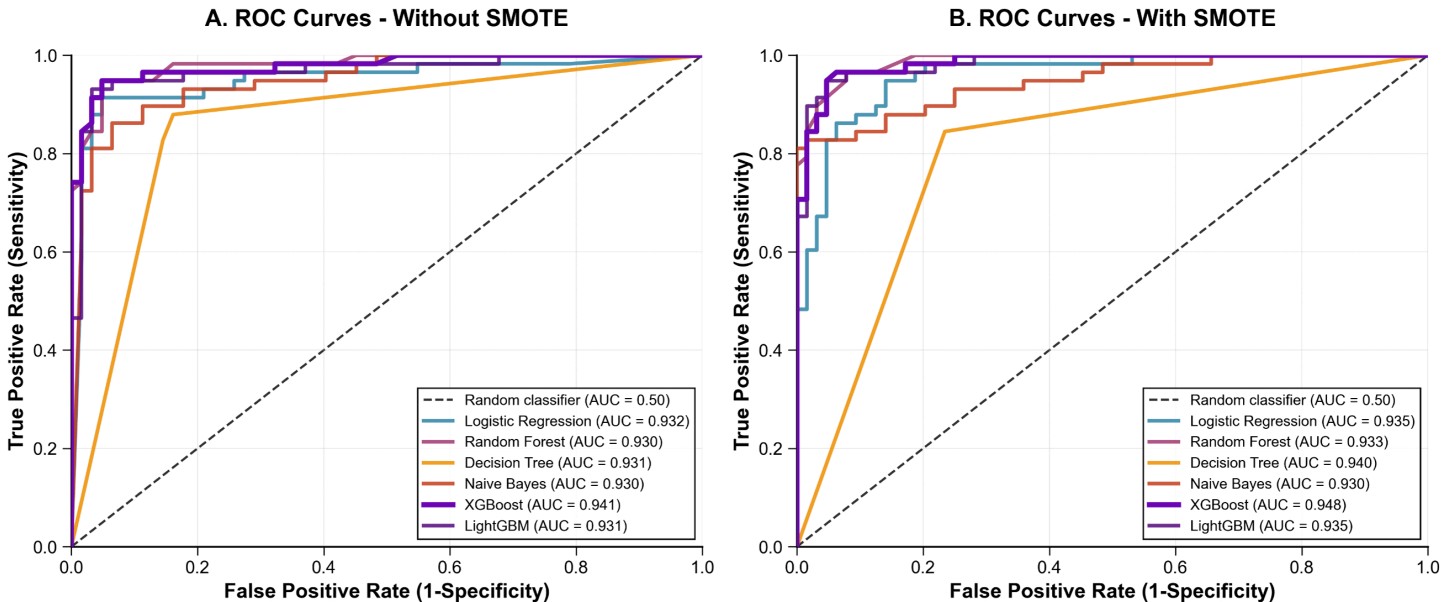

**Fig 3**. **ROC curve comparison for Dataset 1 (UAE Tawam Hospital).** Panel A: without SMOTE; Panel B: with SMOTE. XGBoost demonstrates improved discrimination with SMOTE (AUC: 0.886 → 0.904).

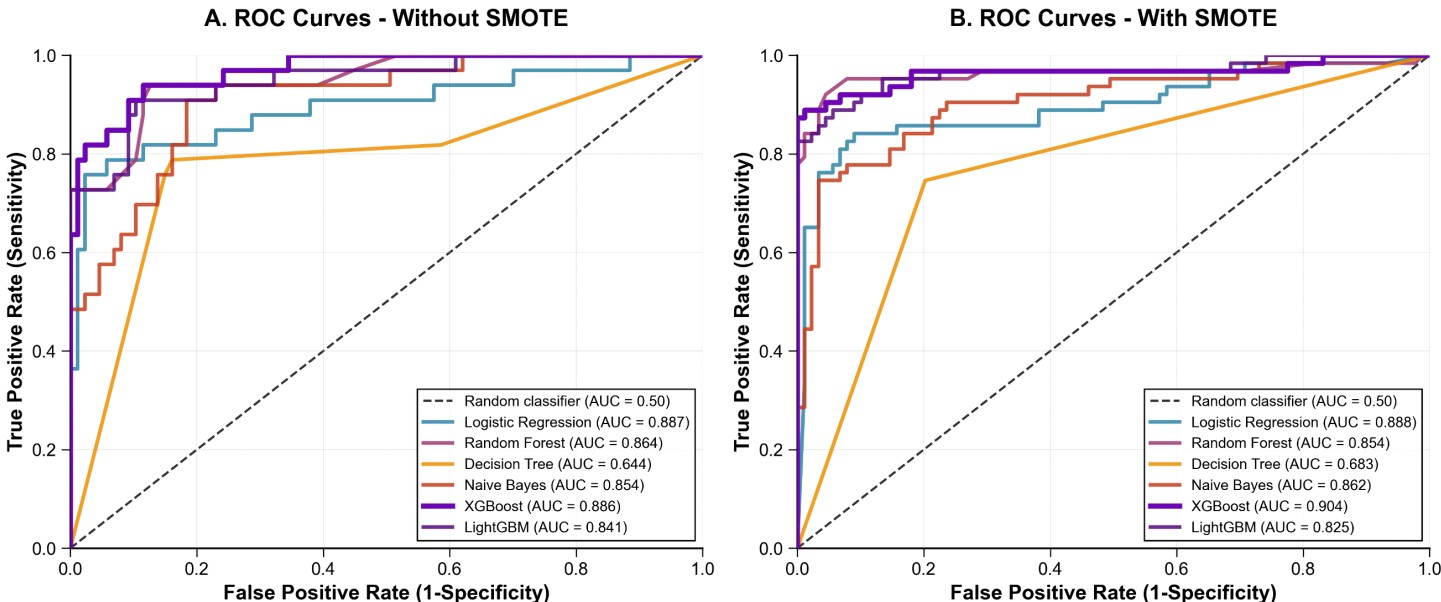

**Fig 4**. **ROC curve comparison for Dataset 2 (UCI CKD).** Panel A: without SMOTE; Panel B: with SMOTE. XGBoost achieves optimal performance (AUC = 0.948 ± 0.013) with SMOTE.

Dataset 1's conservative performance aligns with expectations for heterogeneous real-world hospital data, while Dataset 2's superior performance suggests more structured diagnostic scenarios.

The consistent ranking of XGBoost across both datasets, combined with appropriate variance measures (standard deviations of 0.013-0.024), establishes its robustness for CKD prediction tasks. SMOTE impact patterns provide clear guidance: class balancing offers substantial benefits in imbalanced datasets but minimal advantage in naturally balanced scenarios.

**Interpretability Analysis with SHAP and LIME.** To provide comprehensive interpretability for XGBoost with SMOTE, we employed both SHapley Additive exPlanations (SHAP) for global insights and Local Interpretable Model-agnostic Explanations (LIME) for patient-specific explanations, ensuring alignment with established clinical knowledge.

**Cross-Dataset Feature Importance Analysis.** The comparative SHAP analysis reveals dataset-specific but clinically coherent feature hierarchies across different clinical contexts.

The comparative SHAP analysis shown in Fig 5 reveals dataset-specific but clinically coherent feature hierarchies. Dataset 1 prioritizes eGFRBaseline, HbA1c, and CholesterolBaseline, reflecting the complex cardiovascular-renal risk profile of hospital-based populations. This hierarchy aligns with known pathophysiology where reduced eGFR indicates declining kidney function, while elevated HbA1c suggests diabetic nephropathy risk.

Dataset 2 emphasizes specific gravity and hemoglobin as primary predictors, focusing on direct renal function indicators and hematological complications typical of CKD progression. This divergence reflects different clinical contexts: the structured UCI dataset captures clear diagnostic patterns, while the real-world hospital dataset encompasses broader risk factor interactions.

**Feature Impact Patterns and Clinical Validation.** The detailed feature impact distributions are illustrated in Fig 6, which demonstrate consistent clinical logic across datasets through SHAP summary plots, validating our conservative optimization approach.

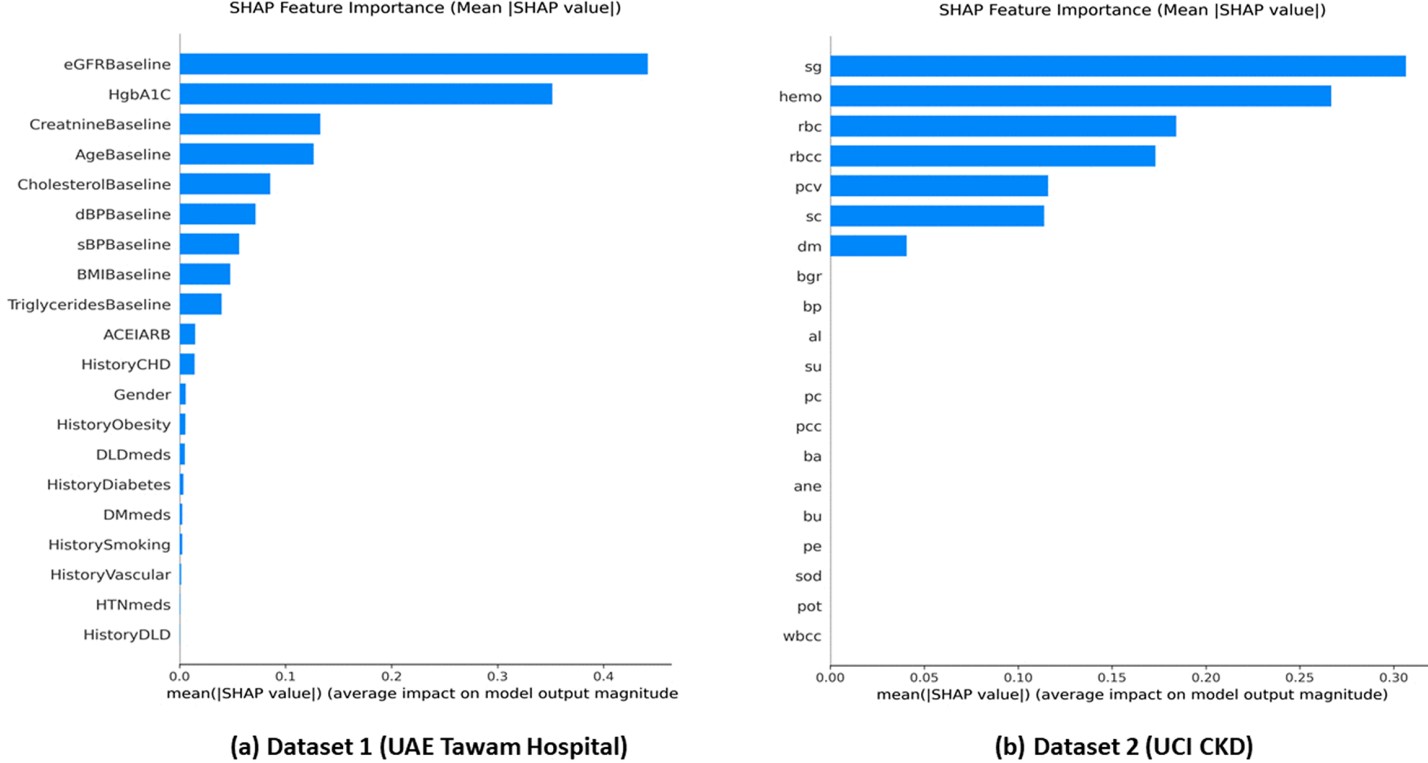

**(a) Dataset 1 (UAE Tawam Hospital)**    **(b) Dataset 2 (UCI CKD)**

**Fig 5**. **SHAP feature importance comparison across datasets.** Panel A: Dataset 1 (UAE Tawam Hospital), where cardiovascular–renal markers are predominantly influential. Panel B: Dataset 2 (UCI CKD), emphasizing direct renal function indicators.

The summary plots presented in Fig 6 demonstrate consistent clinical logic across datasets. In Dataset 1, low eGFR values strongly increase CKD risk, consistent with clinical guidelines where eGFR < 60 mL/min/1.73m² indicates CKD. High HbA1c levels contribute to CKD prediction, reflecting diabetes-related kidney damage when HbA1c > 7%.

Dataset 2 shows similar clinically appropriate patterns: low specific gravity and low hemoglobin values drive positive CKD predictions, while normal values provide strong protection. The scatter distributions reveal clinically meaningful thresholds that align with established diagnostic criteria.

**Individual Prediction Transparency.** Patient-level interpretability is exemplified in Fig 7, which provides transparent clinical reasoning through SHAP waterfall plots for representative cases from both datasets.

The waterfall analyses demonstrated in Fig 7 show how SHAP enables transparent patient-level clinical reasoning. Both examples show patients correctly classified as non-CKD through interpretable decision processes. The Dataset 1 patient benefits from excellent eGFR (103.6 mL/min/1.73m²) providing strong protection, while the Dataset 2 patient shows normal hemoglobin and specific gravity values providing protection against CKD classification.

**Patient-Level Interpretability with LIME.** To complement SHAP's global insights, we applied LIME to generate case-specific rationales for representative XGBoost predictions across both datasets (Figs 8 and 9). LIME constructs simple surrogate models locally around individual predictions by perturbing feature values and observing prediction changes.

**Case A (Sample 50 - UCI Dataset):** The model correctly predicted CKD classification for a patient with multiple renal dysfunction indicators. Consistent with SHAP global analysis, LIME reveals low specific gravity (sg ≤ 1.01) as the strongest CKD risk factor, followed by severe anemia (hemo ≤ 11.20 g/dL). Additional contributors include elevated serum creatinine, presence of red blood cells in urine, and low red blood cell count.

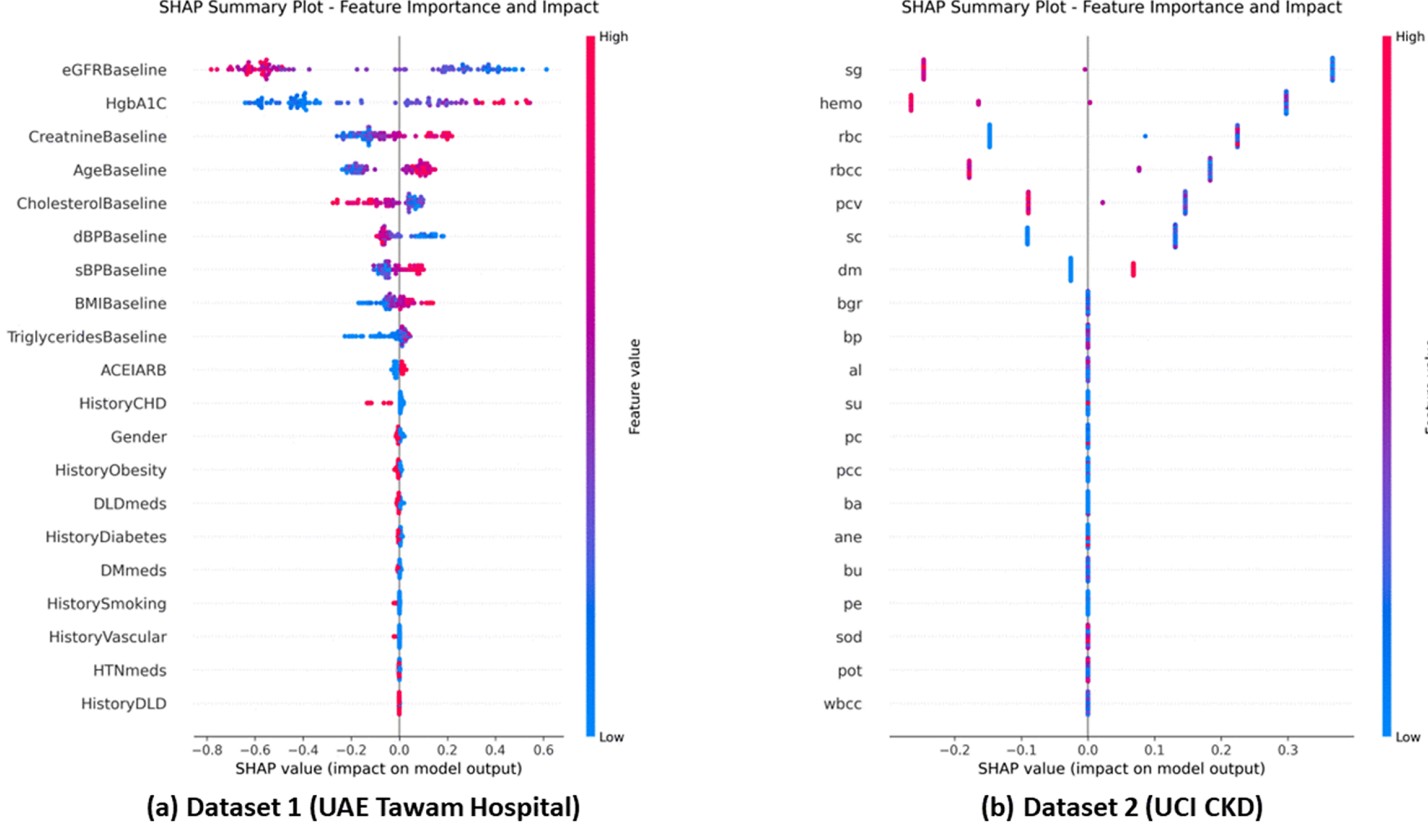

**Fig 6**. **SHAP summary plot comparison across datasets.** Panel A: Dataset 1 (UAE Tawam Hospital). Panel B: Dataset 2 (UCI CKD). Feature value distributions highlight clinically coherent patterns and consistent model behavior across both cohorts.

**Case B (Sample 49 - Tawam Dataset):** The model correctly predicted non-CKD classification for a patient with excellent renal function despite some cardiovascular risk factors. Consistent with SHAP's identification of eGFRBaseline as most important, LIME reveals excellent kidney function (eGFRBaseline > 101.63 mL/min/1.73m²) as the dominant protective factor, along with optimal glycemic control (HbA1c ≤ 5.90%) and normal creatinine levels.

**SHAP-LIME Convergence and Clinical Deployment.** The convergence between SHAP global patterns and LIME individual explanations validates model interpretability across clinical contexts. Both methods consistently identify the same clinically relevant biomarkers: direct renal and hematological markers (specific gravity, hemoglobin) in Dataset 2, and cardiovascular-renal markers (eGFR, HbA1c) in Dataset 1.

This dual interpretability approach facilitates clinical workflow integration through EHR systems that can display both global risk factor hierarchies and patient-specific explanations in real-time. The consistency between SHAP and LIME outputs enables automated risk alerts with transparent reasoning, personalized intervention recommendations, and enhanced patient communication through clear explanations that connect individual risk status to broader clinical knowledge.

The result is a clinically actionable framework that balances evidence-based population health insights with personalized medicine approaches, supporting informed clinical decision-making in CKD prevention and management while providing the interpretability foundation necessary for successful clinical deployment.

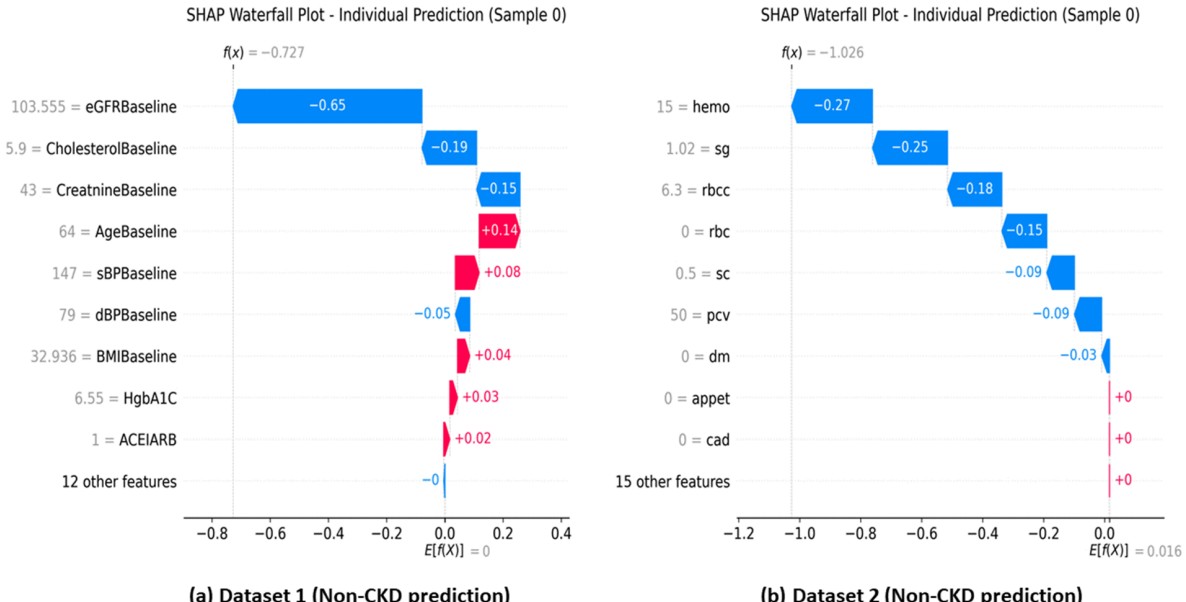

**(a) Dataset 1 (Non-CKD prediction)**   **(b) Dataset 2 (Non-CKD prediction)**

**Fig 7**. **Individual prediction explanations using SHAP waterfall plots.** Panel A: Dataset 1, representative *Non-CKD* prediction. Panel B: Dataset 2, representative *Non-CKD* prediction. The plots provide transparent, case-level clinical reasoning by showing how each feature contribution shifts the model output from the baseline toward the final decision.

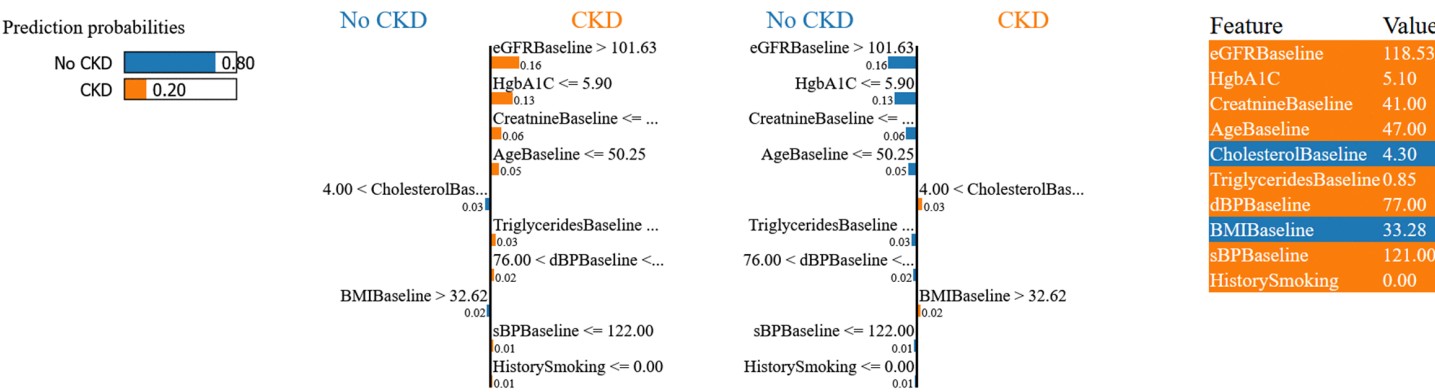

**Fig 8**. **LIME explanation for UCI Dataset case.** Sample 50 showing CKD prediction driven by low specific gravity and severe anemia, validating SHAP's hematological marker hierarchy.

### 5.3 Model calibration analysis

**Context and baselines.** To contextualize calibration, we compare Brier scores to a base-rate classifier that predicts the cohort prevalence $\pi$ for all samples: $\text{Brier}_{\text{base}} = \pi(1-\pi)$. Dataset 1 (Tawam) has $\pi \approx 0.114$ ($\text{Brier}_{\text{base}} \approx 0.10$); Dataset 2 (UCI) has $\pi \approx 0.625$ ($\text{Brier}_{\text{base}} \approx 0.234$).

**Dataset 1 (Tawam).** Without SMOTE, calibration was good for LR (Brier 0.076; ECE 0.018), RF (0.078; 0.017), and XGB (0.080; 0.030). LightGBM was moderate (0.093; 0.063), while DT and NB were less well calibrated (0.128/0.134; 0.114/0.123). With SMOTE, calibration degraded as expected (e.g., XGB 0.112/0.129; LR 0.130/0.179). Fig 10 summarizes Brier/ECE, and Table 7 reports the corresponding numerical values for all models and settings; when reliable probabilities are required, post-hoc recalibration (e.g., isotonic or Platt scaling) is advisable.

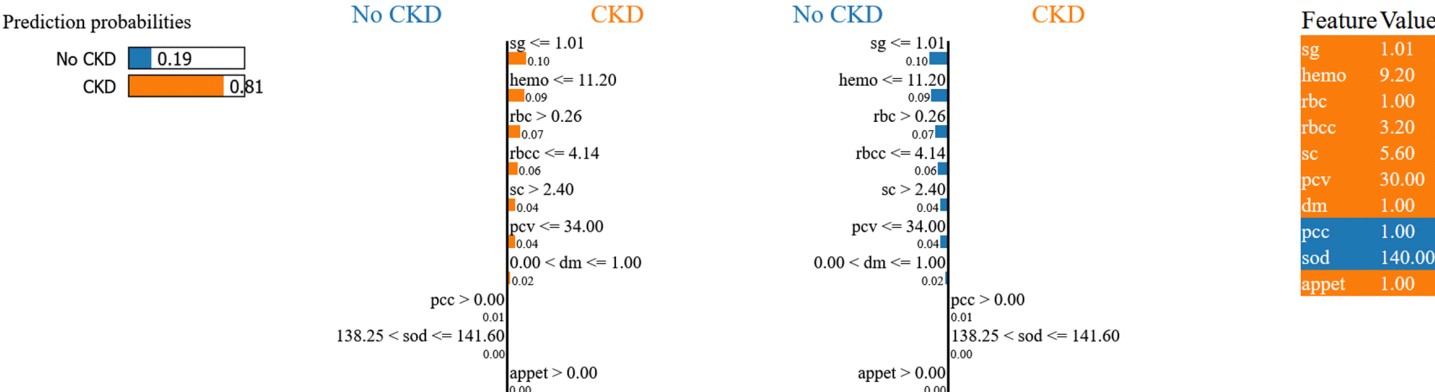

**Fig 9. LIME explanation for Tawam Dataset case.** Sample 49 showing non-CKD prediction dominated by excellent eGFR protection, consistent with SHAP's cardiovascular-renal emphasis.

**Table 7**. **Calibration metrics (Brier, ECE) from outer-fold OOF predictions (lower is better).**

| Model (setting) | Brier (Tawam) | ECE (Tawam) | Brier (UCI) | ECE (UCI) |
|---|---|---|---|---|
| Logistic Regression (no SMOTE) | **0.076** | 0.018 | 0.030 | **0.009** |
| Random Forest (no SMOTE) | 0.078 | **0.017** | 0.011 | 0.027 |
| Decision Tree (no SMOTE) | 0.128 | 0.114 | 0.017 | 0.011 |
| Naive Bayes (no SMOTE) | 0.134 | 0.123 | 0.108 | 0.124 |
| XGBoost (no SMOTE) | 0.080 | 0.030 | 0.013 | 0.045 |
| LightGBM (no SMOTE) | 0.093 | 0.063 | **0.008** | 0.012 |
| Logistic Regression (SMOTE) | 0.130 | 0.179 | 0.030 | 0.015 |
| Random Forest (SMOTE) | 0.100 | 0.111 | 0.010 | 0.023 |
| Decision Tree (SMOTE) | 0.159 | 0.164 | 0.019 | 0.015 |
| Naive Bayes (SMOTE) | 0.184 | 0.193 | 0.104 | 0.118 |
| XGBoost (SMOTE) | 0.112 | 0.129 | 0.010 | 0.028 |
| LightGBM (SMOTE) | 0.098 | 0.083 | 0.013 | 0.013 |

Values computed from outer-fold out-of-fold (OOF) predictions of the 10-fold outer loop. Bold indicates the best (lowest) value in each column. Tawam = Dataset 1; UCI = Dataset 2.

**Dataset 2 (UCI).** Calibration was excellent and far below the base-rate Brier, with minimal SMOTE effects: LGBM (0.008; 0.012), RF (0.011; 0.027), DT (0.017; 0.011), XGB (0.013; 0.045), LR (0.030; 0.009). NB was less well calibrated (0.108; 0.124). Fig 11 shows reliability curves that remain close to the identity line, while Table 7 provides Brier and ECE values.

## 6 Discussion

This study demonstrates the feasibility of integrating interpretable machine learning with chronic kidney disease (CKD) prediction across diverse clinical contexts. Our findings establish XGBoost with SHAP and LIME explainability as an effective framework for transparent CKD risk assessment while addressing the fundamental barrier of algorithmic "black boxes" in healthcare applications.

### 6.1 Cross-domain validation of interpretable ML

The primary contribution of this work lies in demonstrating that explainable AI techniques maintain clinical coherence across fundamentally different healthcare contexts. Dataset 1 (UAE Tawam Hospital) represents complex

Summary of Calibration (Brier & ECE)

Metric = Brier · Metric = ECE

Fig 10. **Calibration summary for Dataset 1 (Tawam): Brier (left) and ECE (right) with/without SMOTE.**

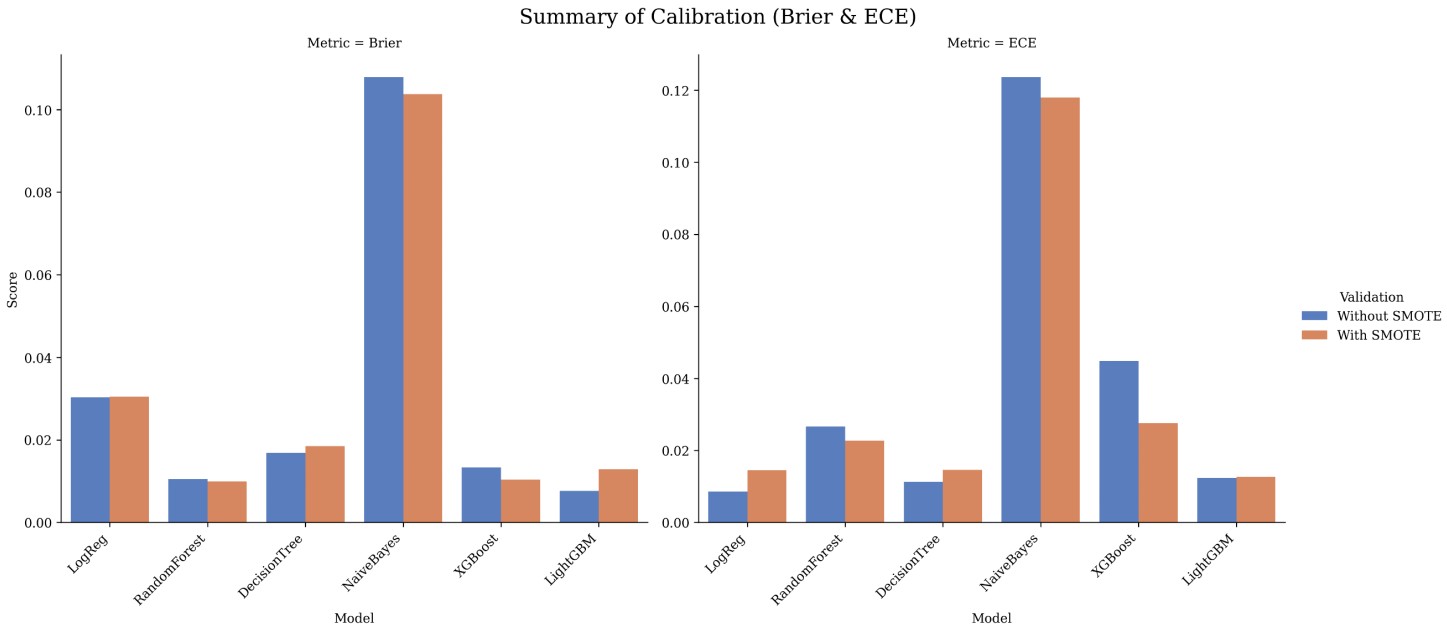

Summary of Calibration (Brier & ECE)

Metric = Brier · Metric = ECE

Fig 11. **Calibration summary for Dataset 2 (UCI): Brier (left) and ECE (right) with/without SMOTE.**

cardiovascular-renal comorbidity scenarios typical of hospital-based populations, while Dataset 2 (UCI CKD) captures structured
diagnostic environments with direct renal markers. Despite these contextual differences, both SHAP global explanations and LIME individual predictions consistently identified clinically appropriate biomarkers.

In the hospital context, eGFRBaseline, HbA1c, and CholesterolBaseline emerged as primary predictors, reflecting the complex pathophysiology where declining kidney function interacts with metabolic dysregulation and cardiovascular risk factors. In the diagnostic context, specific gravity, hemoglobin, and serum creatinine dominated predictions, emphasizing direct renal function assessment and CKD-related anemia. This contextual adaptation validates that interpretable ML frameworks can provide meaningful insights regardless of clinical setting complexity.

The convergence between SHAP population-level patterns and LIME patient-specific explanations across both datasets strengthens confidence in model reliability. This consistency indicates that XGBoost learned clinically appropriate decision rules that operate coherently from systematic screening protocols to individualized patient care, essential for real-world clinical deployment.

## 6.2 Clinical performance and SMOTE impact

Performance differences between datasets (88.4% vs 94.6% accuracy) reflect realistic expectations for clinical ML applications rather than algorithmic limitations. Dataset 1's conservative performance aligns with complex real-world medical data characterized by multiple comorbidities and inherent uncertainty. Dataset 2's superior performance suggests clearer diagnostic boundaries in structured clinical scenarios. Importantly, our conservative optimization approach prevented unrealistic near-perfect results that would indicate overfitting.

SMOTE application revealed context-dependent benefits crucial for clinical implementation guidance. In Dataset 1's imbalanced environment, sensitivity improved dramatically from 25.7% to 60.5%, representing substantial clinical value for CKD screening where early detection is paramount. Dataset 2 showed minimal SMOTE impact due to natural class balance, validating that preprocessing strategies must be tailored to specific healthcare environments rather than applied uniformly.

Consistent with Sect 5.3, SMOTE substantially improved sensitivity on Dataset 1 while degrading probability calibration (higher Brier/ECE), indicating that post-hoc recalibration may be required when calibrated risk estimates are needed. By contrast, Dataset 2 maintained excellent calibration with minimal SMOTE effect.

## 6.3 Interpretability framework advantages

Our dual SHAP-LIME approach addresses complementary clinical needs. SHAP provides population-level insights for evidence-based protocol development and clinical guideline formation. LIME enables patient-specific explanations that support bedside decision-making and patient communication. The perfect alignment between these global and local explanations across diverse clinical contexts demonstrates the framework's robustness for clinical deployment.

Feature importance hierarchies align consistently with established CKD pathophysiology, validating model clinical relevance. Low eGFR values driving CKD predictions correspond to clinical guidelines where eGFR < 60 mL/min/1.73m² indicates disease presence. Elevated HbA1c contributions reflect diabetes-related nephropathy mechanisms. Similarly, low specific gravity and hemoglobin patterns indicate impaired renal concentrating ability and CKD-related anemia, respectively.

## 6.4 Comparative performance and clinical credibility

While previous studies reported higher raw accuracy values (95-96.5%) on similar datasets [43,44], these approaches lack comprehensive interpretability frameworks essential for clinical adoption. Our 94.6% accuracy combined with transparent explanations provides superior clinical utility compared to "black box" high-performance models. Studies reporting perfect accuracy [45] likely suffer from overfitting artifacts that limit real-world applicability.

Our conservative optimization strategy deliberately maintains clinically credible performance levels that reflect inherent uncertainty in medical prediction tasks. This approach prioritizes generalizability and clinical trust over maximizing training metrics, addressing healthcare providers' legitimate concerns about algorithmic reliability.

## 6.5 Clinical implementation considerations

Computational efficiency (< 0.1 seconds per prediction) ensures real-time decision support compatible with clinical workflows and EHR integration. Feature importance insights provide actionable guidance: hospital-based settings should emphasize eGFR monitoring and glycemic control, while diagnostic contexts should focus on urinalysis and hemoglobin assessment.

However, implementation must address interpretability method limitations. LIME's local linear approximations require interpretation alongside SHAP global insights and established clinical evidence. SHAP's feature independence assumptions may not capture complex physiological interactions. Continuous model monitoring and potential recalibration remain essential as patient populations and healthcare systems evolve.

## 6.6 Implications for precision medicine

This work advances the integration of trustworthy AI into clinical practice by demonstrating that high-performance machine learning can deliver transparent, interpretable predictions **while preserving competitive discrimination**. The successful use of interpretable frameworks across distinct clinical contexts provides a roadmap for implementing explainable AI in other medical domains while preserving the clinical reasoning processes essential for patient safety and provider confidence.

Our framework addresses a key transparency barrier that has limited ML adoption in healthcare, offering a validated path for advancing precision medicine in nephrology through interpretable artificial intelligence.

## 6.7 Limitations and future research directions

Several limitations warrant consideration. First, the modest sample sizes (n=491 and n=400) and the single-center nature of the Tawam Hospital cohort may limit generalizability. Second, as demonstrated in our calibration analysis (Sect 5.3), class balancing with SMOTE improves sensitivity but degrades probability calibration, which may necessitate site-specific post-hoc recalibration. Third, the structural differences between our datasets (cardio-renal/metabolic versus direct renal diagnostic contexts) limit the feasibility of an informative harmonized-subset external validation; we therefore flag this as a priority for prospective multi-site work. Fourth, residual confounding from unmeasured socioeconomic and genetic factors may persist. Finally, our study focused on established XAI methods (SHAP for global, LIME for local interpretability); the absence of newer XAI paradigms (e.g., counterfactual or concept-based explanations) represents an avenue for future work.

Future research should prioritize:

1. external validation across multi-center longitudinal datasets to establish generalizability and temporal stability;
2. integration of additional data modalities (genetic, environmental, behavioral) for enhanced precision;
3. development of dynamic prediction models that adapt to evolving patient trajectories; and
4. systematic audits of model fairness across demographic groups and cost-effectiveness studies compared with standard protocols.

## 6.8 Clinical impact and implementation

This study establishes a robust framework for interpretable CKD prediction that integrates XGBoost performance with SHAP and LIME explainability across diverse clinical contexts. Our approach addresses the transparency barrier to ML adoption by showing that high-performance models can maintain clinical interpretability while delivering clinically meaningful performance.

The framework provides actionable guidance for deployment through: (1) dataset-specific optimization strategies that recognize varying class-balancing benefits across clinical environments; (2) feature-importance hierarchies aligned with

established clinical knowledge that augment rather than replace clinical expertise; and (3) convergent global and local explanations that support both systematic protocols and individualized care decisions.

Implementation considerations include EHR integration using healthcare standards (e.g., HL7 FHIR), real-time decision support capabilities, and comprehensive training programs for appropriate model interpretation. Regulatory compliance, algorithmic fairness, and patient engagement strategies remain essential for successful deployment.

**In conclusion**, interpretable machine learning can enhance CKD prediction while maintaining clinical transparency, providing a validated foundation for advancing precision medicine in nephrology and supporting trustworthy AI integration into diverse healthcare workflows.

## 7 Conclusion

This study establishes an interpretable machine learning framework for chronic kidney disease (CKD) prediction that successfully integrates XGBoost performance with SHAP and LIME explainability across diverse clinical datasets. XGBoost with SMOTE optimization achieved 88.4% accuracy (AUC = 0.904) on the real-world UAE Tawam Hospital dataset and 94.6% accuracy (AUC = 0.948) on the UCI CKD dataset, with conservative optimization preventing overfitting artifacts commonly reported in medical machine learning.

The interpretability analysis revealed clinically coherent feature hierarchies aligning with established CKD pathophysiology. The UAE dataset prioritized cardiovascular-renal markers (eGFRBaseline, HbA1c, CholesterolBaseline), while the UCI dataset emphasized direct renal indicators (specific gravity, hemoglobin, serum creatinine). The convergence between SHAP global patterns and LIME individual explanations validates model reliability across population and patient-level decision-making.

Key limitations include structural differences between datasets limiting cross-validation, potential synthetic data artifacts from SMOTE application, and the need for external validation across diverse healthcare systems. However, our framework addresses the fundamental barrier to machine learning adoption in healthcare by providing transparent predictions without sacrificing accuracy.

Future research should prioritize multi-center external validation, integration of additional clinical variables, and clinical implementation studies evaluating real-world effectiveness compared to standard care protocols. This work demonstrates that interpretable machine learning can enhance CKD prediction while maintaining clinical transparency, providing a foundation for advancing precision medicine in nephrology and supporting trustworthy artificial intelligence integration into healthcare practice.

## Acknowledgments

The authors thank Tawam Hospital (UAE) for providing anonymized clinical data and acknowledge the UCI Machine Learning Repository. This work was supported by the RAISS Laboratory, Department of Mathematics and Computer Science, INPT, Rabat, Morocco.

## Author contributions

**Conceptualization:** El Mehdi Chouit, Mohamed Rachdi, Brahim Raouyane.

**Data curation:** El Mehdi Chouit, Brahim Raouyane.

**Formal analysis:** El Mehdi Chouit, Mohamed Rachdi, Brahim Raouyane.

**Funding acquisition:** El Mehdi Chouit.

**Methodology:** El Mehdi Chouit, Mohamed Rachdi, Mostafa Bellafkih, Brahim Raouyane.

**Project administration:** Mostafa Bellafkih, Brahim Raouyane.

**Software:** El Mehdi Chouit, Mohamed Rachdi.

**Supervision:** Mostafa Bellafkih, Brahim Raouyane.

**Validation:** El Mehdi Chouit, Mohamed Rachdi, Mostafa Bellafkih, Brahim Raouyane.

**Visualization:** El Mehdi Chouit, Brahim Raouyane.

**Writing – original draft:** El Mehdi Chouit, Mostafa Bellafkih, Brahim Raouyane.

**Writing – review & editing:** Mostafa Bellafkih, Brahim Raouyane.

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
