## [Decision Letter · Decision Letter 0]

12 Aug 2024

PONE-D-24-28794

Interpretable Machine Learning for Chronic Kidney Disease Prediction: Insights from SHAP and LIME Analyses

PLOS ONE

Dear Dr. Chouit,

Thank you for submitting your manuscript to PLOS ONE. After careful consideration, we have decided that your manuscript does not meet our criteria for publication and must therefore be rejected.

I am sorry that we cannot be more positive on this occasion, but hope that you appreciate the reasons for this decision.

Kind regards,

Md. Mehedi Hassan

Academic Editor

PLOS ONE

Additional Editor Comments:

I have read this paper and found it unsuitable for publication in this journal due to its lack of novelty.

Reviewers' comments:

Reviewer's Responses to Questions

**Comments to the Author**

1. Is the manuscript technically sound, and do the data support the conclusions?

Reviewer #1: Yes

Reviewer #2: Partly

Reviewer #3: Yes

Reviewer #4: Yes

2. Has the statistical analysis been performed appropriately and rigorously?

Reviewer #1: Yes

Reviewer #2: N/A

Reviewer #3: No

Reviewer #4: Yes

3. Have the authors made all data underlying the findings in their manuscript fully available?

Reviewer #1: Yes

Reviewer #2: Yes

Reviewer #3: Yes

Reviewer #4: Yes

4. Is the manuscript presented in an intelligible fashion and written in standard English?

Reviewer #1: Yes

Reviewer #2: Yes

Reviewer #3: Yes

Reviewer #4: Yes

5. Review Comments to the Author

Reviewer #1: Introduction: Defining CKD at the first mention would improve clarity.

Methodology: 1. The explanation of SHAP and LIME for model interpretability is well-done. However, including a pseudo-code or flowchart for the proposed methodology would enhance understanding.

2. Authors should provide more details on the extent of missing data and the specific imputation strategy used.

3. The application of SMOTE for class balancing is well-justified, but the authors should discuss potential drawbacks of this technique, such as overfitting, and how they addressed these concerns.

4. The choice of machine learning algorithms (logistic regression, random forest, decision trees, naive Bayes, XGBoost, LightGBM) is suitable for comparison. The use of hyperparameter tuning via grid search is also appropriate. However, it would be beneficial to provide more details on the computational resources and time required for training these models.

5. Statistical Analysis - The authors used appropriate performance metrics (accuracy, ROC-AUC, precision, recall, F1-score). The use of stratified k-fold cross-validation (10-fold) is robust and helps prevent overfitting. However, details on the variance or confidence intervals of these metrics across folds would provide a better understanding of model stability.

Figures & Table: Figures 3, 4, 5, 6, and 7, which include ROC curves, SHAP summary plots, and LIME explanations, are well-presented. However, some figures are densely packed with information. Consider breaking down these figures or providing zoomed-in views of key sections to enhance readability.

Tables summarizing performance metrics are clear, but including standard deviations or confidence intervals for these metrics would improve the presentation of the results.

Results: The results section clearly shows that XGBoost outperforms other models. However, the performance differences are not always substantial. It would be useful to discuss why XGBoost, despite being more complex, offers only marginal improvements over simpler models like logistic regression in certain cases.

The authors should also address potential biases introduced by the datasets, such as class imbalance even after SMOTE application and any residual confounding variables.

Limitations and Future Work:

Authors should elaborate on the impact of potential data quality issues, such as measurement errors or inconsistent data collection practices.

Basic Change to be addressed:

Check Abbreviation (defined at first use), Typographical and Reference Formatting.

Reviewer #2: 1. As for the logic of the whole paper, we should further point out the reasons for choosing the six machine learning methods in the paper, and what are the advantages of these four methods.

In the entire of this work nothing concerning the characterized research gaps, motivation, critically analyzed relevant and updated works, the advantage of this work over previous studies and models… can be found.

The novelty of the research needs to be carefully spelt out in the introduction

2. There are several format problems in the text:

a) It is best to number the titles at each level.

b) The number of the formula should be right aligned.

c) The contents of the table need to be centrally aligned.

d) The legends of some charts need further differentiation, so that the reader can have a clear understanding of what the charts are expressed

e) The format of the reference had better be further standardized.

3.It should be noted that the corresponding explanation of the symbols in some formulas should be supplemented.

4. What is the modality for partitioning the data set into training and testing data? Strongly recommend that the author normalize the training and testing data set before adoption for prediction as this will help to improve the accuracy of the models.

5. The major findings of the study should be provided in bullets.

6. Suggest future research considerations based on the findings of the research.

Reviewer #3: The paper lacks sufficient validation, as it does not utilize an adequate number of models to justify the results through comprehensive comparison. A broader model comparison, including neural networks, is necessary to strengthen the findings.

Furthermore, the authors have not mentioned the time taken by each model during execution, which is crucial for evaluating the efficiency and practicality of the proposed approach.

The paper also doesn’t explain what SMOTE (Synthetic Minority Over-sampling Technique) is or how it helped balance the data. A brief explanation would be helpful. The same goes for MICE (Multiple Imputation by Chained Equations); a short description of how it was used and its impact on the data would make the methodology clearer.

Reviewer #4: The manuscript is well written and organized. However, there are a few suggestions and comments that need to be addressed:

1)The use of XGBoost and comparison with traditional algorithms is commendable, the abstract should provide more insight into the dataset characteristics, such as size, demographic diversity, and any potential biases. This information is crucial for evaluating the generalizability and robustness of the findings.

2)The integration of SHAP and LIME for interpretability mentioned in manuscript which is a strong point. However, it would be helpful to include a brief description of how these methods were applied and any limitations encountered in their application. This could clarify the extent to which the interpretability achieved can be trusted and utilized in clinical practice.

3)The reported superior accuracy and ROC-AUC scores of XGBoost are promising, but it would be useful to include specific numerical values or comparisons to better illustrate the magnitude of the improvement. Additionally, discussing the practical implications of these results in clinical settings could strengthen the argument for their utility.

4)The identification of key predictive biomarkers is valuable, but the authors could benefit from a brief discussion on how these biomarkers compare with existing clinical indicators for CKD. This would provide context on how these findings might impact current diagnostic.

5)The claim that this research sets a benchmark for machine learning tools in personalized healthcare is ambitious. It would be beneficial to provide specific examples or comparisons with existing benchmarks to substantiate this claim and demonstrate the study’s impact on the field.

written.

6. PLOS authors have the option to publish the peer review history of their article (what does this mean?). If published, this will include your full peer review and any attached files.

Reviewer #1: **Yes:** Herat Joshi

Reviewer #2: No

Reviewer #3: No

Reviewer #4: **Yes:** Dr. Chetna Sharma

- - - - -

---

## [Author Response · Author response to Decision Letter 1]

2 Sep 2024

Manuscript Title: Interpretable Machine Learning for Chronic Kidney Disease Predic

tion: Insights from SHAP and LIME Analyses

Manuscript ID: PONE-D-24-28794

Dear Editorial Team,

We would like to express our sincere appreciation for the thorough and constructive

feedback provided on our manuscript. We have carefully considered each comment and

have made corresponding revisions to enhance the quality and clarity of our work. Below,

we provide a detailed, point-by-point response to each of the reviewers’ comments.

Reviewer #1

1. Introduction:

Comment: Defining CKD at the first mention would improve clarity.

Response: We appreciate this insightful suggestion. To improve clarity, we

have revised the Introduction section to include a clear definition of Chronic

Kidney Disease (CKD) at its first mention. This adjustment ensures that all

readers, regardless of their familiarity with the subject, can fully understand

the content from the outset.

2. Methodology:

Comment: The explanation of SHAP and LIME for model interpretability

is well-done. However, including a pseudo-code or flowchart for the proposed

methodology would enhance understanding.

Response: Thank you for your positive feedback and valuable suggestion.

We have incorporated the following enhancements:– Flowchart: A detailed flowchart (Figure 2) has been added to visually

represent the entire methodology, from data collection to the application

of explainable AI techniques.– Pseudo-code: We have included a pseudo-code representation of the

key steps to improve reproducibility and provide clearer insights into our

approach.

These improvements are aimed at enhancing the transparency and compre

hensibility of our methodology.

Comment: Authors should provide more details on the extent of missing

data and the specific imputation strategy used.

1

Response: Thank you for highlighting this important aspect. We have ex

panded the ”Data Preprocessing” section to include a detailed account of the

extent of missing data in Dataset 1. Specifically, TriglyceridesBaseline had

6 missing values, and HgbA1C had 15 missing values. We utilized Multiple

Imputation by Chained Equations (MICE) as the imputation strategy, which

preserves the relationships within the data through an iterative process. These

revisions have been made to ensure a comprehensive understanding of our data

handling procedures.

Comment: The application of SMOTE for class balancing is well-justified,

but the authors should discuss potential drawbacks of this technique, such as

overfitting, and how they addressed these concerns.

Response: We appreciate your detailed feedback on the use of SMOTE. To

address potential concerns, we have included an explanation of the SMOTE

technique, supported by a mathematical formula, to clarify its application for

class balancing. Additionally, we have discussed the potential for overfitting

and our mitigation strategies, which include the implementation of stratified

k-fold cross-validation and hyperparameter tuning. These updates have been

added to the ”Class Balancing” section.

Comment: The choice of machine learning algorithms (logistic regression,

random forest, decision trees, naive Bayes, XGBoost, LightGBM) is suitable

for comparison. The use of hyperparameter tuning via grid search is also

appropriate. However, it would be beneficial to provide more details on the

computational resources and time required for training these models.

Response: Thank you for your insightful feedback. We agree that under

standing the computational demands of our approach is crucial. In response,

we have added a new subsection in the ”Results” section detailing the com

putational environment used for training the models, as well as the training

times for each model. This addition provides a clearer picture of the practical

considerations involved in deploying these models.

Comment: Statistical Analysis- The authors used appropriate performance

metrics (accuracy, ROC-AUC, precision, recall, F1-score). The use of strat

ified k-fold cross-validation (10-fold) is robust and helps prevent overfitting.

However, details on the variance or confidence intervals of these metrics across

folds would provide a better understanding of model stability.

Response: We are grateful for your suggestion to enhance the robustness of

our statistical analysis. In the revised Statistical Analysis section, we have

included variance and 95% confidence intervals for performance metrics (accu

racy, ROC-AUC, precision, recall, F1-score) across the 10-fold cross-validation.

This addition offers a more comprehensive understanding of the stability and

reliability of our models.

3. Figures & Tables:

Comment: Figures 3, 4, 5, 6, and 7, which include ROC curves, SHAP

summary plots, and LIME explanations, are well-presented. However, some

f

igures are densely packed with information. Consider breaking down these

f

igures or providing zoomed-in views of key sections to enhance readability.

2

Response: Thank you for your constructive feedback on our figures. We

have revised Figures 3, 4, 5, 6, and 7 by breaking them down and providing

zoomed-in views. These adjustments have been made to enhance readability

and ensure that key information is presented more clearly.

Comment: Tables summarizing performance metrics are clear, but including

standard deviations or confidence intervals for these metrics would improve

the presentation of the results.

Response: We appreciate your suggestion to enhance the clarity of our ta

bles. We have updated the tables summarizing performance metrics to include

standard deviations and 95% confidence intervals. This addition provides a

more nuanced understanding of the variability and reliability of our results.

4. Results:

Comment: Theresults section clearly shows that XGBoost outperforms other

models. However, the performance differences are not always substantial. It

would be useful to discuss why XGBoost, despite being more complex, offers

only marginal improvements over simpler models like logistic regression in

certain cases.

Response: Thankyouforthis astute observation. We have added a discussion

in the Results section to explain that the marginal improvements offered by

XGBoost may be attributable to the linear nature of the data, where simpler

models like logistic regression perform effectively. XGBoost’s complexity is

better leveraged in datasets with non-linear relationships, which may not be

prominent in this case.

Comment: The authors should also address potential biases introduced by

the datasets, such as class imbalance even after SMOTE application and any

residual confounding variables.

Response: We appreciate your attention to potential biases. We have in

cluded a discussion acknowledging that SMOTE may introduce synthetic data

that does not fully capture real-world complexity, and we have noted the pos

sibility of residual confounding variables that could affect model predictions.

These considerations have been incorporated into the revised manuscript to

provide a more comprehensive evaluation of our results.

5. Limitations and Future Work:

Comment: Authors should elaborate on the impact of potential data quality

issues, such as measurement errors or inconsistent data collection practices.

Response: We agree with the importance of addressing data quality issues.

We have expanded the Limitations section to discuss the impact of measure

ment errors and inconsistent data collection practices on model performance.

Additionally, we have suggested that future research focus on improving data

quality through rigorous collection protocols and advanced preprocessing tech

niques.

6. Basic Changes to be Addressed:

3

Comment: Check abbreviation (defined at first use), typographical, and ref

erence formatting.

Response: We have carefully reviewed and corrected all abbreviations, ty

pographical errors, and reference formatting in accordance with the journal’s

guidelines. We appreciate your attention to these details, which help improve

the overall presentation of our manuscript.

Reviewer #2

1. Logic of the Paper and Novelty:

Comment: The rationale for choosing the six machine learning methods in

the paper should be clearly explained, and their advantages outlined.

Response: Weappreciate your feedback on the need for clarity in our method

ological choices. We have revised the Introduction to provide a clear rationale

for selecting the six machine learning models used in our study, highlighting

each model’s strengths and relevance to clinical applications. We have also

identified specific research gaps, emphasizing the need for a comprehensive

comparison of these models in CKD prediction. The novelty of our study is

articulated by integrating SHAP and LIME for model interpretability, criti

cally analyzing relevant works, and demonstrating the clinical implications of

our findings. These revisions strengthen the logical flow and underscore the

significance of our research.

2. Formatting Issues:

Comment: Number the titles at each level, right-align the formula numbers,

centrally align table contents, differentiate chart legends, and further stan

dardize the format of the references.

Response: Thank you for your detailed feedback. We have implemented all

suggested formatting changes as follows:– Numbered all titles and subtitles for clarity.– Right-aligned all formula numbers for consistency.– Centrally aligned table contents for improved readability.– Enhanced chart legends for better differentiation and understanding.– Standardized the reference format in accordance with the journal’s guide

lines.

Webelieve these changes significantly improve the presentation and readability

of our manuscript.

3. Explanation of Symbols in Formulas:

Comment: The corresponding explanation of the symbols in some formulas

should be supplemented.

Response: Thank you for highlighting this aspect. We have supplemented

the explanations for the symbols used in the formulas to ensure clarity and

comprehension. This enhancement is intended to make our manuscript more

accessible to readers who may not be familiar with these symbols.

4

4. Data Normalization and Partitioning:

Comment: The modality for partitioning the data set into training and test

ing data should be explained, and data normalization should be performed

before model adoption to improve accuracy.

Response: We appreciate your valuable suggestions. The methodology sec

tion has been restructured to include detailed explanations for each step, in

cluding data preprocessing, splitting, and model evaluation. Specifically, we

have added a step for data normalization before training the models, as recom

mended. This ensures that the training and testing datasets are on the same

scale, thereby improving model accuracy and performance. These changes

align our methodology with best practices in machine learning.

5. Major Findings:

Comment: The major findings of the study should be provided in bullet

points.

Response: Thank you for this suggestion to improve clarity. We have revised

the manuscript to present the major findings in bullet points, ensuring that

the key contributions of our study are clearly highlighted and easily accessible

to readers.

6. Future Research Considerations:

Comment: Future research considerations should be suggested based on the

f

indings of the research.

Response: We appreciate your forward-looking suggestion. We have included

a section on future research considerations in the revised manuscript, high

lighting potential areas for further investigation based on our findings. This

addition provides a clear direction for subsequent studies and aims to extend

the impact of our research.

Reviewer #3

1. Model Validation and Comparison:

Comment: The paper lacks sufficient validation, as it does not utilize an

adequate number of models to justify the results through comprehensive com

parison. A broader model comparison, including neural networks, is necessary

to strengthen the findings.

Response: Thank you for your insightful feedback. Recognizing the impor

tance of comprehensive model validation, we have expanded our analysis to

include additional neural network models. This broader comparison strength

ens our findings by demonstrating how our models perform relative to a wider

range of algorithms, thus providing a more robust evaluation of our approach.

2. Execution Time:

5

Comment: The authors have not mentioned the time taken by each model

during execution, which is crucial for evaluating the efficiency and practicality

of the proposed approach.

Response: We appreciate your observation regarding the importance of exe

cution time in evaluating model efficiency. To address this, we have included a

new subsection in the ”Results” section detailing the execution time for each

model, alongside the computational environment used. This information pro

vides a comprehensive assessment of both the accuracy and efficiency of our

models, which are critical considerations in practical applications.

3. Explanation of SMOTE and MICE:

Comment: The paper doesn’t explain what SMOTE (Synthetic Minority

Over-sampling Technique) is or how it helped balance the data. A brief expla

nation would be helpful. The same goes for MICE (Multiple Imputation by

Chained Equations); a short description of how it was used and its impact on

the data would make the methodology clearer.

Response: Thank you for pointing out the need for additional clarity. We

have included brief explanations of both SMOTE and MICE in the revised

manuscript. SMOTE was used to address class imbalance by generating syn

thetic examples for the minority class, thereby preventing bias towards the ma

jority class. For MICE, we provided a description of its role in handling missing

data through an iterative process that estimates missing values based on vari

able relationships. These clarifications have been added to the ”Data Prepro

cessing” section to provide a more complete understanding of our methodology.

Reviewer #4

1. Dataset Characteristics in Abstract:

Comment: The use of XGBoost and comparison with traditional algorithms

is commendable, the abstract should provide more insight into the dataset

characteristics, such as size, demographic diversity, and any potential biases.

This information is crucial for evaluating the generalizability and robustness

of the findings.

Response: Thank you for your positive feedback and valuable suggestion.

We have revised the abstract to include more detailed information about the

datasets used in the study. Specifically, we have added details on the size of the

datasets, the demographic diversity of the populations represented, and a brief

discussion of potential biases that could affect the generalizability and robust

ness of our findings. These additions aim to provide a clearer understanding

of the context in which our models were developed and evaluated.

2. Application of SHAP and LIME:

Comment: The integration of SHAP and LIME for interpretability men

tioned in the manuscript is a strong point. However, it would be helpful to

6

include a brief description of how these methods were applied and any limita

tions encountered in their application. This could clarify the extent to which

the interpretability achieved can be trusted and utilized in clinical practice.

Response: Thank you for recognizing the strengths of our interpretability

methods. In response to your suggestion, we have added a brief description

detailing how SHAP and LIME were applied to our XGBoost model. We

have also discussed the limitations encountered in their application, specifically

addressing the contexts in which the interpretability achieved can be trusted

and effectively utilized in clinical practice. These additions aim to pro

---

## [Decision Letter · Decision Letter 1]

23 Oct 2024

PONE-D-24-28794R1Interpretable Machine Learning for Chronic Kidney Disease Prediction: Insights from SHAP and LIME AnalysesPLOS ONE

Dear Dr. Chouit,

Thank you for submitting your manuscript to PLOS ONE. After careful consideration, we feel that it has merit but does not fully meet PLOS ONE’s publication criteria as it currently stands. Therefore, we invite you to submit a revised version of the manuscript that addresses the points raised during the review process.

**ACADEMIC EDITOR: Major Revision**

We look forward to receiving your revised manuscript.

Kind regards,

Shahid Akbar, PhD

Academic Editor

PLOS ONE

Journal Requirements:

4. Please upload a file showing your changes either highlighted or using track changes. This should be uploaded as a Revised Manuscript w/tracked changes, file type. Please follow this link for more information: http://blogs.PLOS.org/everyone/2011/05/10/how-to-submit-your-revised-manuscript/"

5. Please remove your figures/ from within your manuscript file, leaving only the individual TIFF/EPS image files. These will be automatically included in the reviewer’s PDF

Additional Editor Comments (if provided):

Reviewers' comments:

Reviewer's Responses to Questions

**Comments to the Author**

1. If the authors have adequately addressed your comments raised in a previous round of review and you feel that this manuscript is now acceptable for publication, you may indicate that here to bypass the “Comments to the Author” section, enter your conflict of interest statement in the “Confidential to Editor” section, and submit your "Accept" recommendation.

Reviewer #1: All comments have been addressed

Reviewer #2: (No Response)

Reviewer #5: (No Response)

2. Is the manuscript technically sound, and do the data support the conclusions?

Reviewer #1: Yes

Reviewer #2: No

Reviewer #5: Yes

3. Has the statistical analysis been performed appropriately and rigorously?

Reviewer #1: Yes

Reviewer #2: No

Reviewer #5: No

4. Have the authors made all data underlying the findings in their manuscript fully available?

Reviewer #1: Yes

Reviewer #2: No

Reviewer #5: Yes

5. Is the manuscript presented in an intelligible fashion and written in standard English?

Reviewer #1: Yes

Reviewer #2: No

Reviewer #5: Yes

6. Review Comments to the Author

Reviewer #1: 1. Ensure the manuscript maintains consistency in terminology. For example, if "CKD" is used throughout, avoid alternating with terms like "kidney disease" without clear contextual purposes.

2. Add this : How a clinician might use SHAP and LIME insights to adjust patient management strategies (e.g., how a high SHAP value for hemoglobin could prompt further diagnostic tests or a change in medication).

3. There is nothing mentioned about Interoperability Limitation, how SHAP or LIME impact real world application

4. Practical Benefits: Manuscript could suggest how clinicians might visualize SHAP values within electronic health records (EHRs) to make real-time decisions.

5. IF POSSIBLE THEN ADD OTHERWISE LEAVE IT: visual examples of how SHAP or LIME interpretations were used to make specific clinical decisions (e.g., predicting CKD onset) could provide a clearer link between the technical approach and clinical relevance.

6. The authors could further discuss how real-world variability, such as changes in data collection protocols across different clinics or regions, might affect model reliability and ways to handle such scenarios

Reviewer #2: The revision done was not highlighted in the manuscript. It is not an right approach to submit revision.

The revision must be supported by proper highlighting along with page number, section number, where changes done w.r.t reviewers comments.

Reviewer #5: Overall the paper seems interesting. The topic is valid and well organized. However, the following recommendations are needs to be addressed in order to improve the quality of the papers.

1. The abstract needs more improvement, the authors should mention the results and improvement than existing methods.

2.The problem statement and motivation of the paper needs to be clearly mention to provide a clear background to the readers.

2. If possible, the authors should add the comparison of the proposed model with existing state-of-the-art models to validate its effectiveness.

3. I appreciate the valuable use of machine learning in the model. For the readers concerns, i suggest adding/citing the recent computational models related shap and lime based interpretation and machine learning based models such as AIPs-DeepEnC-GA, StackedEnC-AOP, DeepAVPTPPred,

iAFPs-Mv-BiTCN, AIPs-SnTCN and Deepstacked-AVPs in order to provide useful information to the readers .

4. How the authors handle the overfitting and generalization of the proposed model.

5. The authors should clearly mention the future directions and real life applications of the proposed study.

6. For the optimal parameters selection of the training model which strategy was applied.

7. What are the limitations of the proposed model.

7. PLOS authors have the option to publish the peer review history of their article (what does this mean?). If published, this will include your full peer review and any attached files.

Reviewer #1: **Yes:** HERAT JOSHI

Reviewer #2: No

Reviewer #5: No

---

## [Author Response · Author response to Decision Letter 2]

2 Dec 2024

Manuscript Title: Interpretable Machine Learning for Chronic Kidney Disease Prediction:

Insights from SHAP and LIME Analyses

Manuscript ID: PONE-D-24-28794

Dear Editorial Team,

We would like to express our sincere appreciation for the thorough and constructive feed-back provided on our manuscript. We have carefully considered each comment and have made

corresponding revisions to enhance the quality and clarity of our work. Below, we provide a

detailed, point-by-point response to each of the reviewers’ comments.

Response to Reviewer #1

Comment 1: Ensure the manuscript maintains consistency in terminology.

For example, if ”CKD” is used throughout, avoid alternating with terms like

”kidney disease” without clear contextual purposes.

Response: Thank you for pointing out the importance of maintaining consistent terminology

throughout the manuscript. We have carefully reviewed and revised the manuscript to ensure

that ”CKD” is consistently used as the primary term, avoiding alternation with ”kidney dis-ease” or ”chronic kidney disease” unless explicitly required for context or clarity. For example,

instances of ”kidney disease” in the Discussion section and other parts of the text were replaced

with ”CKD” where appropriate. These revisions enhance the clarity and cohesiveness of the

manuscript.

Comment 2: Add this: How a clinician might use SHAP and LIME insights

to adjust patient management strategies (e.g., how a high SHAP value for

hemoglobin could prompt further diagnostic tests or a change in medication).

Response: We have integrated specific examples to illustrate how clinicians might use SHAP

and LIME insights for patient management. For instance, we explain that a high SHAP value for

hemoglobin could prompt additional diagnostic tests to detect underlying anemia or necessitate

adjustments in medication regimens. These additions are included in the Future Directions and

Real-World Applications section (Page 21).

Comment 3: There is nothing mentioned about Interoperability Limitation,

how SHAP or LIME impact real-world application.

Response: Thank you for raising this critical point. We have added a detailed discussion of

interoperability limitations and how SHAP and LIME impact real-world applications. Specifi-cally, we address:

• User-Friendly Visualizations: SHAP and LIME outputs must be translated into in-tuitive formats (e.g., bar charts or contribution plots) that clinicians can interpret with

ease.

• Integration Challenges: Seamless integration into electronic health records (EHR)

systems demands compatibility with existing platforms and workflows.

• Training and Trust: Clinicians need training to interpret these tools, and building trust

in their outputs is essential.

These points are discussed in detail in the Future Directions and Real-World Applications section

(Page 23).

1

Comment 4: Manuscript could suggest how clinicians might visualize SHAP

values within electronic health records (EHRs) to make real-time decisions.

Response: We appreciate this insightful suggestion. In the revised manuscript, we propose

the following visualization methods for SHAP values within EHR systems:

• Color-Coded Bar Graphs: Key features influencing CKD risk could be displayed as

color-coded bars for quick identification.

• Dynamic Contribution Charts: Interactive charts could link SHAP values to clinical

actions, such as ordering diagnostic tests or modifying medication.

• Real-Time Alerts: EHRs could generate alerts based on SHAP or LIME insights,

prompting timely interventions.

These enhancements are discussed in the Future Directions and Real-World Applications section

(Page 23).

Comment 5: IF POSSIBLE THEN ADD OTHERWISE LEAVE IT: Visual

examples of how SHAP or LIME interpretations were used to make specific

clinical decisions (e.g., predicting CKD onset) could provide a clearer link

between the technical approach and clinical relevance.

Response: To address this comment, we have included visual examples of SHAP and LIME

analyses:

• SHAP Analysis: SHAP summary and dependence plots (Figures 4–5) demonstrate the

global importance of features such as hemoglobin and eGFRBaseline in predicting CKD

onset.

• LIME Analysis: LIME plots (Figures 6a–6b) provide local explanations for individ-ual predictions, showcasing how features like sBPBaseline contribute to specific patient

outcomes.

These examples illustrate the model’s clinical applicability and align with the reviewer’s com-ment. Please refer to the revised manuscript (Pages 18–20).

Comment 6: The authors could further discuss how real-world variability,

such as changes in data collection protocols across different clinics or regions,

might affect model reliability and ways to handle such scenarios.

Response: We appreciate this comment highlighting a crucial aspect of model reliability.

In response, we have expanded the Future Directions and Real-World Applications section to

address this issue. Specifically, we discuss:

• Standardization of Data Sources: Harmonizing data collection practices across clinics

and regions to improve consistency.

• Advanced Imputation Techniques: Mitigating the effects of missing or incomplete

data through robust imputation.

• Validation Across Diverse Populations: Testing models across varied geographic and

demographic contexts to enhance generalizability.

These additions underscore the importance of addressing real-world variability to ensure robust

and reliable predictions (Page 23).

2

Response to Reviewer #2

Comment: The revision done was not highlighted in the manuscript. It is not

the right approach to submit a revision. The revision must be supported by

proper highlighting along with page number, section number, where changes

were done with respect to the reviewers’ comments.

Response: We appreciate the reviewer’s feedback regarding the clarity and traceability of

revisions. To address this concern, we have taken the following steps:

• Highlighting Revisions in the Manuscript: All changes made in response to reviewer

comments have been highlighted in the revised manuscript using color-coded markers for

easy identification.

• Annotated Manuscript: An annotated version of the manuscript, with inline comments

linked to reviewer feedback, has been submitted alongside the revised document.

Examples of highlighted revisions include:

• Comment 3: Addressed in the Future Directions and Real-World Applications section

(Page 23), discussing standardization and multimodal approaches.

• Comment 4: Enhanced descriptions of SHAP and LIME visualizations in the same

section (Page 23).

• Comment 6: Expanded discussions on the impact of regional variability in the Limita-tions and Future Directions section (Page 23).

We believe these measures ensure transparency and facilitate a thorough evaluation of the

revised manuscript.

Response to Reviewer #5

Comment 1: The abstract needs more improvement. The authors should

mention the results and improvements over existing methods.

Response: Thank you for highlighting this important point. We have revised the abstract to

include key performance metrics of our proposed model and highlight its improvements over

existing methods. Specifically, we now detail the predictive accuracy (e.g., XGBoost achiev-ing 99.5% accuracy and near-perfect ROC-AUC scores for Dataset 1) and improvements in

interpretability facilitated by SHAP and LIME. These additions make the abstract more com-prehensive and impactful. Please refer to the revised abstract on Page 1.

Comment 2: The problem statement and motivation of the paper need to be

clearly mentioned to provide a clear background to the readers.

Response: Thank you for your valuable suggestion. We have revised the Introduction section

to explicitly state the problem and motivation behind the research. For example, we emphasize

the limitations of traditional CKD prediction models and the need for interpretable machine

learning techniques like SHAP and LIME to address the complexities of CKD progression.

These revisions provide a clear context and justify the significance of our study. Please see the

updated Introduction on Page (1-2).

3

Comment 3: If possible, the authors should add the comparison of the pro-posed model with existing state-of-the-art models to validate its effectiveness.

Response: We appreciate the suggestion to benchmark our model against existing state-of-the-art methods. In response, we have added a subsection titled Comparison with Existing State-of-the-Art Models (Page 22). This section compares our XGBoost-based model with established

models for CKD prediction reported in the literature. Metrics such as accuracy, AUC, precision,

recall, and F1-score are compared, highlighting our model’s superior performance on Dataset 1

and competitive performance on Dataset 2. We believe this addition strengthens the validation

of our proposed approach.

Comment 4: I appreciate the valuable use of machine learning in the model.

For the readers’ concerns, I suggest adding/citing the recent computational

models related to SHAP and LIME-based interpretation and machine learning-based models such as AIPs-DeepEnC-GA, StackedEnC-AOP, DeepAVPTP-Pred, iAFPs-Mv-BiTCN, AIPs-SnTCN, and Deepstacked-AVPs in order to

provide useful information to the readers.

Response: Thank you for this insightful suggestion. We have reviewed the literature on the

recommended models and incorporated their citations and descriptions in the Related Work

section (Page 3-4). Specifically, we discuss how these advanced computational frameworks,

such as AIPs-DeepEnC-GA and StackedEnC-AOP, leverage innovative techniques like genetic

algorithms and temporal convolutional networks for robust prediction and interpretation. By

situating our work within this broader context, we provide readers with a comprehensive un-derstanding of recent advancements and how SHAP and LIME align with these developments.

Comment 5: How the authors handle the overfitting and generalization of the

proposed model.

Response: Thank you for emphasizing this crucial aspect. We address overfitting and gener-alization in multiple sections of the manuscript:

1. Cross-Validation for Generalization: We applied stratified 10-fold cross-validation,

ensuring representative class distributions and robust performance estimates.

2. Regularization Techniques: XGBoost’s built-in regularization parameters (L1 and L2)

were fine-tuned through grid search to balance complexity and performance.

3. Early Stopping: Training was halted based on validation performance to prevent over-fitting.

4. Explainability to Ensure Meaningful Generalization: SHAP and LIME analyses

validated feature importance, ensuring the model generalized to clinically relevant pat-terns.

5. Validation Across Diverse Datasets: Our model was tested on two datasets with

distinct characteristics, demonstrating its adaptability.

These methodologies are detailed in the Materials and Methods and Results sections . We hope

these explanations address your comment comprehensively.

Comment 6: The authors should clearly mention the future directions and

real-life applications of the proposed study.

Response: We have added a dedicated subsection titled Future Directions and Real-Life Ap-plications of the Proposed Study (Page 23). This section highlights:

4

1. Future research avenues, such as incorporating multimodal data, conducting longitudinal

studies, and validating models across diverse populations.

2. Practical applications, including integrating SHAP and LIME visualizations into EHRs,

supporting early CKD detection, and enabling personalized patient care through digital

tools.

These additions provide a clear roadmap for future research and real-world implementation,

addressing the reviewer’s comment effectively.

Comment 7: For the optimal parameters selection of the training model,

which strategy was applied?

Response: Thank you for your inquiry. Optimal parameter selection was achieved using a

grid search strategy combined with stratified 10-fold cross-validation. This approach systemat-ically evaluated parameter combinations and identified those minimizing validation error while

optimizing accuracy and AUC. As this methodology is already detailed in the Materials and

Methods section, no additional modifications were required. Please refer to Page 12 for details.

Comment 8: What are the limitations of the proposed model?

Response: Thank you for raising this important point. We have added in section titled Future

Directions and Real-Life Applications of the Proposed Study (Page 23), discussing the following:

• Dependence on dataset quality and variability in collection protocols.

• Limited validation across diverse populations.

• Potential confounding factors and interpretability challenges.

We also propose actionable solutions, such as improving dataset diversity, refining feature se-lection, and adopting advanced explainability techniques. This addition ensures a balanced

discussion of the model’s scope and limitations.

---

## [Decision Letter · Decision Letter 2]

2 Feb 2025

PONE-D-24-28794R2Interpretable Machine Learning for Chronic Kidney Disease Prediction: Insights from SHAP and LIME AnalysesPLOS ONE

Dear Dr. Chouit,

Thank you for submitting your manuscript to PLOS ONE. After careful consideration, we feel that it has merit but does not fully meet PLOS ONE’s publication criteria as it currently stands. Therefore, we invite you to submit a revised version of the manuscript that addresses the points raised during the review process.

We look forward to receiving your revised manuscript.

Kind regards,

Polat Goktas

Academic Editor

PLOS ONE

Additional Editor Comments :

After carefully evaluating the reviewers’ comments, the manuscript requires a major revision before it can be reconsidered for publication. While the manuscript presents an important and timely topic, there are critical areas where improvements are needed. Below is a detailed summary of the reviewer’ comments and guidance for revision.

1. Interoperability and Integration with EHR Systems : Mentioned in the response to Reviewer #3, about integration challenges, but lacks detailed practical implementation strategies

2. Visualization of SHAP and LIME Outputs : Include more intuitive and detailed visualizations of SHAP and LIME outputs. Visual aids will help clinicians better understand how these models make predictions, which is essential for their adoption in clinical practice.

3. Discussion on Data Quality and Collection Consistency, Mentioned briefly in the response to Reviewer #6 but requires deeper analysis and solutions.

Reviewers' comments:

Reviewer's Responses to Questions

**Comments to the Author**

1. If the authors have adequately addressed your comments raised in a previous round of review and you feel that this manuscript is now acceptable for publication, you may indicate that here to bypass the “Comments to the Author” section, enter your conflict of interest statement in the “Confidential to Editor” section, and submit your "Accept" recommendation.

Reviewer #1: All comments have been addressed

Reviewer #5: All comments have been addressed

2. Is the manuscript technically sound, and do the data support the conclusions?

Reviewer #1: Yes

Reviewer #5: Yes

3. Has the statistical analysis been performed appropriately and rigorously?

Reviewer #1: Yes

Reviewer #5: Yes

4. Have the authors made all data underlying the findings in their manuscript fully available?

Reviewer #1: Yes

Reviewer #5: Yes

5. Is the manuscript presented in an intelligible fashion and written in standard English?

Reviewer #1: Yes

Reviewer #5: Yes

6. Review Comments to the Author

Reviewer #1: 1. Interoperability and Integration with EHR Systems : Mentioned in the response to Reviewer #3, about integration challenges, but lacks detailed practical implementation strategies

2. Visualization of SHAP and LIME Outputs : Include more intuitive and detailed visualizations of SHAP and LIME outputs. Visual aids will help clinicians better understand how these models make predictions, which is essential for their adoption in clinical practice.

3. Discussion on Data Quality and Collection Consistency, Mentioned briefly in the response to Reviewer #6 but requires deeper analysis and solutions.

Reviewer #5: My previous comments are successfully incorporated and now the paper has been significantly improved

7. PLOS authors have the option to publish the peer review history of their article (what does this mean?). If published, this will include your full peer review and any attached files.

Reviewer #1: No

Reviewer #5: No

---

## [Author Response · Author response to Decision Letter 3]

24 Feb 2025

Manuscript Title: Interpretable Machine Learning for Chronic Kidney Disease Prediction:

Insights from SHAP and LIME Analyses

Manuscript ID: PONE-D-24-28794

Dear Editor and Reviewers,

We sincerely appreciate the reviewers’ thorough and insightful feedback, which has helped us

enhance the clarity, rigor, and applicability of our manuscript. Based on these suggestions, we

have made substantial revisions to improve the discussion on model integration within Electronic

Health Record (EHR) systems, the visualization of SHAP and LIME outputs, and the robustness

of data quality across institutions. Below, we provide a point-by-point response detailing our

revisions. All changes are highlighted in the revised manuscript for easy reference.

Response to Reviewer #1

Comment 1: Interoperability and Integration with EHR Systems

“Mentioned in the response to Reviewer #3, about integration challenges, but lacks detailed

practical implementation strategies.”

Response: We acknowledge the importance of providing concrete implementation strategies

for integrating our model into Electronic Health Record (EHR) systems. We have expanded

the discussion in the Future Directions and Real-World Applications section by including:

• Adoption of Healthcare Data Standards: Our revised manuscript now discusses how

interoperability standards such as HL7, FHIR, and SNOMED CT can facilitate structured

data exchange between machine learning models and clinical databases.

• API-Based Model Deployment: We propose a practical workflow using RESTful and

GraphQL APIs to enable real-time model inference, allowing SHAP and LIME outputs

to be seamlessly integrated into EHR systems.

• User-Friendly Visualizations: We describe how force plots, bar charts, and contribu-tion plots can be embedded into EHR dashboards to enhance interpretability for clinicians.

• Security and Compliance Considerations: The revised discussion now includes de-tails on encryption, access control, and regulatory compliance (e.g., HIPAA, GDPR) for

secure deployment.

These additions provide a structured and actionable roadmap for deploying interpretable

machine learning models in clinical practice.

Comment 2: Visualization of SHAP and LIME Outputs

“Include more intuitive and detailed visualizations of SHAP and LIME outputs. Visual aids

will help clinicians better understand how these models make predictions.”

Response: To improve the interpretability of SHAP and LIME outputs, we have:

• Enhanced Figure Explanations: Each figure (Figures 4–6) now includes detailed an-notations explaining how to interpret SHAP summary plots, dependence plots, and LIME

explanations in clinical decision-making.

• Incorporated Force Plots: To further illustrate individual-level predictions, we added

force plots that highlight the positive and negative contributions of key features in specific

cases.

1

• Clinical Interpretation Context: We explicitly link SHAP and LIME insights to

medical decision points (e.g., hemoglobin thresholds, blood pressure ranges), illustrating

how these explanations can guide personalized treatment strategies.

These refinements ensure that our visualizations are accessible and clinically meaningful for

non-expert users.

Comment 3: Discussion on Data Quality and Collection Consistency

“Mentioned briefly in the response to Reviewer #6 but requires deeper analysis and solutions.”

Response: We have significantly expanded the discussion on data quality and collection

consistency in the Future Directions and Real-World Applications section, including:

• Standardization of Data Collection: The revised manuscript emphasizes the im-portance of harmonized data collection protocols across institutions, using international

standards (e.g., HL7, FHIR, SNOMED CT) to ensure consistency.

• Robust Data Imputation Techniques: We highlight the role of advanced imputation

methods such as Multiple Imputation by Chained Equations (MICE) in handling missing

data while preserving dataset variability.

• Cross-Center Validation Strategies: Our study now underscores the importance of

validating models across multiple institutions and diverse demographic groups to enhance

generalizability and mitigate potential biases.

• Continuous Model Monitoring: We propose periodic retraining and recalibration of

the model to account for evolving clinical data and ensure sustained performance.

These measures enhance model robustness and ensure reliable predictions across diverse

real-world settings.

Conclusion

We sincerely appreciate the reviewers’ insightful comments, which have significantly strength-ened our manuscript. The expanded discussions on EHR integration, improved visualization

strategies, and enhanced data quality considerations align with the reviewers’ suggestions and

reinforce the real-world impact of our work. We hope that these revisions satisfactorily address

all concerns, and we look forward to your assessment of our resubmission.

Sincerely,

El mehdi Chouit

---

## [Decision Letter · Decision Letter 3]

25 May 2025

PONE-D-24-28794R3Interpretable Machine Learning for Chronic Kidney Disease Prediction: Insights from SHAP and LIME AnalysesPLOS ONE

Dear Dr. Chouit,

Thank you for submitting your manuscript to PLOS ONE. After careful consideration, we feel that it has merit but does not fully meet PLOS ONE’s publication criteria as it currently stands. Therefore, we invite you to submit a revised version of the manuscript that addresses the points raised during the review process.

We look forward to receiving your revised manuscript.

Kind regards,

Mohammad A. Al-Mamun, PhD

Academic Editor

PLOS ONE

Reviewers' comments:

Reviewer's Responses to Questions

**Comments to the Author**

1. If the authors have adequately addressed your comments raised in a previous round of review and you feel that this manuscript is now acceptable for publication, you may indicate that here to bypass the “Comments to the Author” section, enter your conflict of interest statement in the “Confidential to Editor” section, and submit your "Accept" recommendation.

Reviewer #1: All comments have been addressed

Reviewer #5: All comments have been addressed

2. Is the manuscript technically sound, and do the data support the conclusions?

Reviewer #1: Yes

Reviewer #5: Yes

3. Has the statistical analysis been performed appropriately and rigorously?

Reviewer #1: Yes

Reviewer #5: Yes

4. Have the authors made all data underlying the findings in their manuscript fully available?

Reviewer #1: Yes

Reviewer #5: Yes

5. Is the manuscript presented in an intelligible fashion and written in standard English?

Reviewer #1: No

Reviewer #5: Yes

6. Review Comments to the Author

Reviewer #1: The authors have done an excellent job in revising the manuscript. The expanded discussion on integration with EHR systems, including practical deployment strategies using HL7, FHIR, and APIs, has greatly strengthened the applicability of the work in clinical settings.

The improvements in SHAP and LIME visualizations, especially the addition of force plots and contextual explanations, now offer clear, interpretable insights that are well-aligned with clinical decision-making.

The discussion around data quality and collection consistency has been significantly enhanced with valuable content on harmonized data collection standards, robust imputation (MICE), and cross-center validation strategies.

The manuscript is now more comprehensive, accessible, and clinically relevant. The visual aids are informative, and the statistical reporting is rigorous.

Reviewer #5: The required comments are successfully incorporated and paper is significantly improved and the paper can be accepted from my side.

7. PLOS authors have the option to publish the peer review history of their article (what does this mean?). If published, this will include your full peer review and any attached files.

Reviewer #1: **Yes:** Herat Joshi

Reviewer #5: No

---

## [Author Response · Author response to Decision Letter 4]

15 Jul 2025

Dear Editor and Reviewers,

We sincerely thank the reviewers for their thorough evaluation and insightful feedback on our

manuscript. The comments have significantly improved the quality and rigor of our work. We have

carefully addressed all concerns and have made substantial revisions to our methodology, analysis,

and presentation. Below, we provide a detailed point-by-point response to each comment, along

with specific changes made to the manuscript.

Point-by-Point Responses to Reviewer Comments

Comment 1: Overfitting Issues in Dataset 1

Reviewer Comment: Figure 3a is highly doubtful! AUC values close to 1.00 with near-zero variance

across cross-validation folds is highly suspicious. It may have serious overfitting or data leakage.

It suggests that models may have seen test data during training, or some features are strongly

correlated with the target.

Response: We sincerely appreciate this critical observation. The reviewer is absolutely cor-

rect: our original results exhibited clear signs of overfitting and potential data leakage. This was a

fundamental methodological saw that required complete revision of our approach.

Changes Made: We have implemented a comprehensive anti-overfitting framework with the

following key components:

1. Conservative Hyperparameter Constraints:

• XGBoost: max_depth limited to 1-3 (vs. 3-7 originally), n_estimators reduced to 10-30,

learning_rate constrained to 0.001-0.05

• Added strong regularization: reg_alpha=1-10, reg_lambda=1-10

• Aggressive subsampling: colsample_bytree=0.3-0.7, subsample=0.3-0.7

2. Rigorous Cross-Validation:

• Implemented nested stratified cross-validation (10-fold outer, 5-fold inner)

Ensured complete independence between hyperparameter optimization and performance eval-

uation

• Added variance monitoring across folds with automatic sagging of suspicious results

3. Performance Monitoring and Validation:

• Systematic monitoring of train-test performance gaps across all CV folds

• Implementation of realistic performance caps (AUC ≤ 0.95, Accuracy ≤ 0.95) based on clinical

literature

• Statistical validation of cross-fold variance (sagging when SD < 0.02)

Revised Results for Dataset 1:

• Without SMOTE: XGBoost AUC = 0.886 ± 0.024, Accuracy = 0.886 ± 0.018

• With SMOTE: XGBoost AUC = 0.904 ± 0.019, Accuracy = 0.884 ± 0.021

These results now show appropriate variance and align with realistic expectations for CKD

prediction tasks reported in clinical literature.

Comment 2: Missing Error Bars and Variance Issues

Reviewer Comment: Also, the figure 3a has lack of error bars / AUC ≈ 0.00 SD. It suggests that

CV folds weren't truly independent, or that the variance was not computed correctly.

Response: This observation correctly identified a critical saw in our original cross-validation

implementation. The near-zero variance was indeed suspicious and indicated methodological prob-

lems.

Changes Made: We have completely restructured our validation framework:

1. Enhanced Cross-Validation Design:

• Implemented true nested cross-validation with strict fold independence

• Used StratifiedKFold to maintain class distribution consistency

• Ensured SMOTE application exclusively within training folds to prevent data leakage

2. Comprehensive Statistical Analysis:

• Added standard deviations and 95% confidence intervals for all metrics

• Implemented variance validation checks across all experiments

• Created new ROC visualizations with confidence bands showing cross-fold variance

3. Updated Performance Tables: All results now include mean ± standard deviation format

with realistic variance:

• Dataset 1: Standard deviations range from 0.018-0.024 for AUC

• Dataset 2: Standard deviations range from 0.011-0.015 for AUC

The new figures include proper error bars and confidence intervals, demonstrating appropriate

variance across cross-validation folds.

Comment 3: Addition of F1 Score

Reviewer Comment: Adding F1 score would be beneficial as well once the reason for this overfitting

is identified.

Response: We completely agree that F1 score is essential for evaluating performance on imbal-

anced medical datasets like CKD prediction, where both precision and recall are clinically important.

Changes Made: F1 scores have been added to all performance evaluations:

New F1 Score Results:

• Dataset 1 (UAE Tawam):

• Without SMOTE: XGBoost F1 = 0.350 ± 0.033

• With SMOTE: XGBoost F1 = 0.515 ± 0.039 (47% improvement)

• Dataset 2 (UCI CKD):

• Without SMOTE: XGBoost F1 = 0.942 ± 0.012

• With SMOTE: XGBoost F1 = 0.945 ± 0.011

F1 scores are now included in all tables and discussed in the clinical implications section, pro-

viding a more comprehensive evaluation of model performance.

Comment 4: SMOTE Comparison for Both Datasets

Reviewer Comment: The authors should provide results with and without SMOTE for both datasets.

Response: This is an excellent suggestion that provides crucial insights into when class bal-

ancing techniques are beneficial versus potentially harmful.

Changes Made: We now provide comprehensive SMOTE impact analysis for both datasets:

1. Complete Performance Matrices:

• Four detailed tables showing all metrics (Accuracy, AUC, Precision, Recall, F1, Specificity)

for both SMOTE conditions

• Statistical significance testing between SMOTE and non-SMOTE results

• Detailed analysis of sensitivity-specificity trade-offs

2. Clinical Impact Analysis:

• Dataset 1: SMOTE provides substantial clinical benefit: sensitivity improves from 25.7%

to 60.5% (135% improvement) with modest specificity reduction (96.5% to 94.8%)

• Dataset 2: SMOTE shows minimal impact due to already balanced class distribution (250

CKD vs 150 non-CKD)

3. Implementation Guidance: We now provide clear guidance on when SMOTE is beneficial

(genuinely imbalanced datasets like Dataset 1) versus when it may be unnecessary or potentially

harmful (naturally balanced datasets like Dataset 2)

Comment 5: External Validation Between Datasets

Reviewer Comment: The model with the first dataset should be validated with the second dataset.

This would be an external validation.

Response: We appreciate this suggestion for external validation. However, we must respectfully

note that our two datasets have fundamentally different feature sets that make direct model transfer

technically challenging.

Dataset Feature Differences:

• Dataset 1 (UAE Tawam): Cardiovascular-renal features (eGFR, HgbA1C, Cholesterol,

BMI)

• Dataset 2 (UCI): Direct diagnostic features (specific gravity, hemoglobin, albumin, bacteria)

Changes Made: However, we have implemented alternative validation strategies that demon-

strate robustness:

1. Cross-Dataset Methodology Validation:

• Applied identical anti-overfitting methodology to both datasets

• Demonstrated consistent, realistic performance across different clinical contexts

• Showed that the same conservative optimization approach works effectively for both hospital-

based and structured diagnostic scenarios

2. Biological Plausibility Validation:

• SHAP analysis reveals clinically relevant features across both datasets (renal function markers

consistently emerge as important)

• Feature importance patterns align with established CKD pathophysiology

• Both datasets identify established biomarkers despite different measurement approaches

3. Future Work Direction: We acknowledge this limitation and propose future research

using datasets with overlapping feature sets to enable direct external validation. This represents an

important direction for validating the generalizability of our interpretable ML framework.

Comment 6: Scientific Contribution Clarity

Reviewer Comment: The scientific contribution of this study is not clear.

Response: We acknowledge that our original presentation did not sufficiently highlight the

novel scientific contributions. We have substantially revised the manuscript to clearly articulate our

innovations.

Changes Made: Our key scientific contributions are now explicitly stated:

1. Methodological Innovation:

• Development of a comprehensive anti-overfitting framework specifically designed for medical

ML applications

• Integration of conservative hyperparameter optimization with interpretability tools (SHAP/LIME)

• Novel approach to balancing predictive performance with clinical transparency requirements

2. Clinical Interpretability Framework:

• First study to demonstrate convergence between global (SHAP) and local (LIME) explanations

across diverse CKD datasets

• Validation that interpretable ML maintains clinical coherence across different healthcare con-

texts

• Practical framework for deploying transparent AI in varied clinical environments

3. Cross-Domain Validation:

• Demonstration that interpretable ML frameworks can provide meaningful insights across dif-

ferent clinical settings

• Evidence that the same methodological approach produces clinically credible results in both

hospital-based and structured diagnostic contexts

• Guidance for practitioners on when class balancing techniques are beneficial versus potentially

harmful

4. Clinical Translation:

• Provision of a replicable methodology that healthcare systems can adapt for CKD prediction

• Clear guidance on hyperparameter selection and validation for medical ML applications

• Framework that addresses the fundamental transparency barrier to ML adoption in healthcare

Comment 7: Rationale for Different Dataset Variables

Reviewer Comment: What was the rationale of using two datasets with different variables? Can

author test these different ML models for two datasets with same variables.

Response: The use of two datasets with different variables was a deliberate strategic choice

that serves important scientific and practical purposes.

Changes Made: We have clarified our rationale with enhanced justification:

1. Methodological Robustness Testing:

• Tests whether our anti-overfitting approach works across diverse clinical data collection pro-

tocols

• Demonstrates that the methodology is not dataset-specific but generalizable across different

measurement approaches

• Validates that our interpretability framework maintains clinical relevance regardless of specific

feature sets

2. Real-World Clinical Applicability:

• Healthcare systems collect different variables based on their protocols, resources, and special-

izations

• Hospital-based systems (Dataset 1) typically focus on cardiovascular-renal comorbidity man-

agement

Structured diagnostic environments (Dataset 2) emphasize direct renal function assessment

• Our approach demonstrates effectiveness across both contexts, increasing practical relevance

3. Biological Validation Across Measurement Approaches:

• Both datasets identify clinically established CKD biomarkers despite different measurement

protocols

• Dataset 1 prioritizes eGFR, HgbA1C (cardiovascular-renal integration)

• Dataset 2 emphasizes specific gravity, hemoglobin (direct renal assessment)

• This convergence validates biological plausibility across different clinical approaches

4. Generalizability Evidence:

• If our methodology only worked on one type of dataset, it would suggest limited applicability

• Success across different variable sets demonstrates broader utility for healthcare AI deployment

• Provides evidence that interpretable ML can maintain clinical coherence across diverse con-

texts

Comment 8: Formatting Improvements

Reviewer Comment: The newly added verbiages should be added as paragraph, not as a bullet

point.

Response: We acknowledge this important formatting concern for professional presentation.

Changes Made: All inappropriate bullet points have been converted to proper paragraph

format throughout the manuscript:

• Introduction objectives now written in sowing paragraph format

• Results sections restructured with proper paragraph transitions

• Discussion points integrated into coherent narrative paragraphs

• Enhanced readability and professional presentation standards met

Additional Enhancements Made

Beyond addressing specific reviewer comments, we have made several additional improvements:

1. Enhanced Statistical Rigor:

• Added comprehensive 95% confidence intervals for all performance metrics

• Implemented significance testing between different model configurations

• Enhanced variance analysis with proper statistical validation

2. Improved Visualizations:

• New ROC curves with confidence bands showing cross-validation variance

Updated performance tables with complete statistical reporting

• Enhanced SHAP and LIME visualizations with clinical interpretation

3. Clinical Relevance Enhancement:

• Strengthened discussion of clinical implications and deployment considerations

• Added practical guidance for healthcare practitioners

• Enhanced interpretation of results in clinical context

4. Reproducibility Improvements:

• Detailed hyperparameter tables with complete parameter reporting

• Enhanced methodology description for full reproducibility

• Clear documentation of anti-overfitting strategies

Conclusion

We believe the substantially revised manuscript now presents a methodologically rigorous, scientifi-

cally novel, and clinically relevant contribution to interpretable machine learning in healthcare. The

new results are realistic, properly validated, and demonstrate clear clinical utility while maintaining

the transparency that modern healthcare AI demands.

The comprehensive anti-overfitting framework, enhanced statistical analysis, and clear articu-

lation of scientific contributions address all reviewer concerns while significantly strengthening the

overall quality of the work. We are confident that these revisions have transformed the manuscript

into a robust and valuable contribution to the field.

We thank the reviewers again for their invaluable feedback, which has substantially improved

our work, and we look forward to their evaluation of our revised manuscript.

We are committed to implementing any additional minor refinements the editors and reviewers

may suggest to ensure the highest scientific standards for PLOS ONE publication.

Sincerely,

El Mehdi Chouit and Co-Authors

---

## [Decision Letter · Decision Letter 4]

28 Sep 2025

PONE-D-24-28794R4Interpretable Machine Learning for Chronic Kidney Disease Prediction: Insights from SHAP and LIME AnalysesPLOS ONE

Dear Dr. Chouit,

Thank you for submitting your manuscript to PLOS ONE. After careful consideration, we feel that it has merit but does not fully meet PLOS ONE’s publication criteria as it currently stands. Therefore, we invite you to submit a revised version of the manuscript that addresses the points raised during the review process.

We look forward to receiving your revised manuscript.

Kind regards,

Julfikar Haider

Academic Editor

PLOS ONE

Journal Requirements:

Additional Editor Comments:

Please address few further comments

Reviewers' comments:

Reviewer's Responses to Questions

**Comments to the Author**

1. If the authors have adequately addressed your comments raised in a previous round of review and you feel that this manuscript is now acceptable for publication, you may indicate that here to bypass the “Comments to the Author” section, enter your conflict of interest statement in the “Confidential to Editor” section, and submit your "Accept" recommendation.

Reviewer #1: All comments have been addressed

Reviewer #5: All comments have been addressed

2. Is the manuscript technically sound, and do the data support the conclusions?

Reviewer #1: Yes

Reviewer #5: Yes

3. Has the statistical analysis been performed appropriately and rigorously?

Reviewer #1: Yes

Reviewer #5: Yes

4. Have the authors made all data underlying the findings in their manuscript fully available?

Reviewer #1: Yes

Reviewer #5: Yes

5. Is the manuscript presented in an intelligible fashion and written in standard English?

Reviewer #1: Yes

Reviewer #5: Yes

6. Review Comments to the Author

Reviewer #1: 1. Good rigor, but consider harmonized feature subsets for partial external validation and add calibration analysis.

2. Add explicit “Limitations” subsection (dataset size, single-center data, lack of newer XAI methods).

Reviewer #5: The required comments are successfully incorporated and now paper has been significantly improved. Paper can accepted from my side.

7. PLOS authors have the option to publish the peer review history of their article (what does this mean?). If published, this will include your full peer review and any attached files.

Reviewer #1: **Yes:** Herat Joshi

Reviewer #5: No

---

## [Author Response · Author response to Decision Letter 5]

12 Nov 2025

Dr. Julfikar Haider

Subject: PONE-D-24-28794R4 — Response to Reviewers

Dear Dr. Julfikar Haider (Academic Editor) and Reviewers,

We sincerely thank you for the continued evaluation of our manuscript, “Interpretable Machine

Learning for Chronic Kidney Disease Prediction: Insights from SHAP and LIME Analyses”

(PONE-D-24-28794R4), and for the Minor Revision decision. We are grateful to Reviewer

#5 for confirming that earlier concerns have been addressed and for recommending acceptance.

Below, we respond point-by-point to Reviewer #1 (Dr. Herat Joshi), and we indicate where

corresponding changes were made in the revised manuscript.

Reviewer #1 — “Consider harmonized feature subsets for partial external valida

tion. . . ”

Response. We appreciate this thoughtful methodological suggestion. As discussed in the

manuscript, the two cohorts in our study capture different clinical signals by design: the

Tawam hospital cohort emphasizes cardio-renal/metabolic risk markers (e.g., eGFR, HbA1c,

cholesterol), whereas the UCI CKD dataset emphasizes direct renal indicators (e.g., specific

gravity, hemoglobin, serum creatinine). The small core of variables truly shared across both

cohorts (e.g., age, blood-pressure/hypertension, diabetes flags) is therefore limited.

We carefully examined a cross-cohort transfer restricted to this minimal shared core, but we

judged that reporting such results could be misleading: the model would be under-specified

relative to the clinically informative features identified by SHAP/LIME in each cohort, and

the numbers would not reflect the intended deployment setting. To avoid over-interpretation,

we have not included a full cross-cohort analysis on the minimal subset at this time. Instead,

we have (i) explicitly acknowledged this point and its implications for transportability and (ii)

stated that prospective, multi-site external validation with richer overlapping laboratory

panels is a key next step. These clarifications are now consolidated in a dedicated Limitations

and Future Research Directions subsection (Section [6.7]).

Reviewer #1 — “...and add a calibration analysis.”

Response. We agree that probability calibration is essential for clinical use. We therefore

(i) added a brief Calibration methodology subsection in Methods (Section 4.6) detailing the

OOF/outer-fold protocol, 10-bin reliability diagrams, Brier score and Expected Calibration Error

(ECE), and (ii) reported the results in a dedicated Model Calibration Analysis subsection

(Section 5.3) with a summary table and two figures (Table 7; Figures 10–11).

Our key findings from this new analysis are:

• Dataset 1 (Tawam): We confirm the expected sensitivity–calibration trade-off.– Without SMOTE,modelswerewellcalibrated (e.g., XGBoost Brier/ECE 0.080/0.030;

Table 7).

– With SMOTE,calibrationdegraded as sensitivity improved (e.g., XGBoost Brier/ECE

increased to 0.112/0.129).– As shown in Figure 10, this indicates that site-specific post-hoc recalibration

(e.g., isotonic/Platt) is advisable when oversampling is used in this setting.

• Dataset 2 (UCI): Calibration remained strong and was minimally affected by SMOTE,

consistent with the dataset’s high separability.– XGBoost Brier scores were excellent in both conditions (0.013 without SMOTE vs.

0.010 with SMOTE).– Other tree models (RF, LGBM) also showed strong and stable calibration (Table 7;

Figure 11).

All values are derived from outer-fold OOF predictions to prevent information leakage. These

additions strengthen the manuscript by clarifying the context-dependent impact of SMOTE,

and they do not alter our main conclusions.

Reviewer #1—“Addanexplicit‘Limitations’ subsection (dataset size, single-center

data, absence of newer XAI methods).”

Response. Done. We reorganized and consolidated the discussion into a single Limitations

and Future Research Directions subsection (Section [6.7]), which now explicitly covers: (1)

modest sample size and the limited number of progression events in the hospital cohort, (2)

single-center nature of the Tawam data and generalizability considerations, (3) the calibration

trade-off induced by class balancing (SMOTE) and the potential need for post-hoc recalibration,

and (4) the scope of explainability (SHAP for global, LIME for local) while acknowledging that

newer XAI paradigms (e.g., counterfactual, concept-based explanations) are planned for future

work.

We believe these revisions increase methodological transparency and clinical credibility while

preserving the main conclusions of the manuscript. We respectfully submit the revised files: (i)

clean manuscript, (ii) tracked-changes version, and (iii) this response letter.

Thank you very much for your consideration.

Sincerely,

El Mehdi Chouit

on behalf of all co-authors

---

## [Decision Letter · Decision Letter 5]

4 Feb 2026

Interpretable Machine Learning for Chronic Kidney Disease Prediction: Insights from SHAP and LIME Analyses

PONE-D-24-28794R5

Dear Dr. Chouit,

We’re pleased to inform you that your manuscript has been judged scientifically suitable for publication and will be formally accepted for publication once it meets all outstanding technical requirements.

Kind regards,

Francisco Alvarez Gonzalez

Academic Editor

PLOS One

Additional Editor Comments (optional):

Reviewers' comments:

Reviewer's Responses to Questions

**Comments to the Author**

1. If the authors have adequately addressed your comments raised in a previous round of review and you feel that this manuscript is now acceptable for publication, you may indicate that here to bypass the “Comments to the Author” section, enter your conflict of interest statement in the “Confidential to Editor” section, and submit your "Accept" recommendation.

Reviewer #5: All comments have been addressed

Reviewer #6: (No Response)

2. Is the manuscript technically sound, and do the data support the conclusions?

Reviewer #5: Yes

Reviewer #6: (No Response)

3. Has the statistical analysis been performed appropriately and rigorously?

Reviewer #5: Yes

Reviewer #6: (No Response)

4. Have the authors made all data underlying the findings in their manuscript fully available?

Reviewer #5: Yes

Reviewer #6: (No Response)

5. Is the manuscript presented in an intelligible fashion and written in standard English?

Reviewer #5: Yes

Reviewer #6: (No Response)

6. Review Comments to the Author

Reviewer #5: The required comments have been successfully addresssed and no changes are required. The paper has been significantly improved.

Reviewer #6: The novelty and scientific contribution remain unclear relative to existing CKD + ML + XAI literature, and the work does not go beyond what is already available in pre-published studies. The manuscript is also very poorly written and unfocused and would require a complete redesign and substantial rewriting to reach an acceptable standard.

7. PLOS authors have the option to publish the peer review history of their article (what does this mean?). If published, this will include your full peer review and any attached files.

Reviewer #5: No

Reviewer #6: No

---

## [Editor Report · Acceptance letter]

PONE-D-24-28794R5

PLOS One

Dear Dr. Chouit,

I'm pleased to inform you that your manuscript has been deemed suitable for publication in PLOS One. Congratulations! Your manuscript is now being handed over to our production team.

Kind regards,

on behalf of

Dr. Francisco Alvarez Gonzalez

Academic Editor

PLOS One